# Phylogenomic analyses of echinoid diversification prompt a re-evaluation of their fossil record

**Nicolás Mongiardino Koch[1,2]\*, Jeffrey R Thompson[3,4], Avery S Hiley[2], Marina F McCowin[2], A Frances Armstrong[5], Simon E Coppard[6], Felipe Aguilera[7], Omri Bronstein[8,9], Andreas Kroh[10], Rich Mooi[5], Greg W Rouse[2]**

[1]Department of Earth & Planetary Sciences, Yale University, New Haven, United States; [2]Scripps Institution of Oceanography, University of California San Diego, La Jolla, United States; [3]Department of Earth Sciences, Natural History Museum, London, United Kingdom; [4]University College London Center for Life's Origins and Evolution, London, United Kingdom; [5]Department of Invertebrate Zoology and Geology, California Academy of Sciences, San Francisco, United States; [6]Bader International Study Centre, Queen's University, Herstmonceux Castle, East Sussex, United Kingdom; [7]Departamento de Bioquímica y Biología Molecular, Facultad de Ciencias Biológicas, Universidad de Concepción, Concepción, Chile; [8]School of Zoology, Faculty of Life Sciences, Tel Aviv University, Tel Aviv, Israel; [9]Steinhardt Museum of Natural History, Tel-Aviv, Israel; [10]Department of Geology and Palaeontology, Natural History Museum Vienna, Vienna, Austria

**\*For correspondence:** nmongiardinokoch@ucsd.edu

**Competing interest:** The authors declare that no competing interests exist.

**Abstract** Echinoids are key components of modern marine ecosystems. Despite a remarkable fossil record, the emergence of their crown group is documented by few specimens of unclear affinities, rendering their early history uncertain. The origin of sand dollars, one of its most distinctive clades, is also unclear due to an unstable phylogenetic context. We employ 18 novel genomes and transcriptomes to build a phylogenomic dataset with a near-complete sampling of major lineages. With it, we revise the phylogeny and divergence times of echinoids, and place their history within the broader context of echinoderm evolution. We also introduce the concept of a chronospace – a multidimensional representation of node ages – and use it to explore methodological decisions involved in time calibrating phylogenies. We find the choice of clock model to have the strongest impact on divergence times, while the use of site-heterogeneous models and alternative node prior distributions show minimal effects. The choice of loci has an intermediate impact, affecting mostly deep Paleozoic nodes, for which clock-like genes recover dates more congruent with fossil evidence. Our results reveal that crown group echinoids originated in the Permian and diversified rapidly in the Triassic, despite the relative lack of fossil evidence for this early diversification. We also clarify the relationships between sand dollars and their close relatives and confidently date their origins to the Cretaceous, implying ghost ranges spanning approximately 50 million years, a remarkable discrepancy with their rich fossil record.

## Editor's evaluation

The study by Mongiardino Koch et al., presents new phylogenomic and molecular clock analyses of echinoids. The study uses state of the art phylogenetic approaches and includes 18 newly sequenced genomes and transcriptomes, which are used to estimate the tree topology and divergence times of major groups of echinoids. The molecular clock-estimated times of origin of

particular echinoid lineages predate the lineages' appearance on the fossil record by tens of millions of years, prompting re-evaluation of the early evolution of echinoid diversity.

## Introduction

The fossil record represents the best source of primary data for constraining the origins of major lineages across the tree of life. However, the fossil record is not perfect, and even for groups with an excellent fossilization potential, constraining their age of origin can be difficult (*Smith and Peterson, 2002*; *Donoghue and Benton, 2007*). Furthermore, as many traditional hypotheses of relationships have been revised in light of large-scale molecular datasets, the affinities of fossil lineages and their bearings on inferred times of divergence have also required a reassessment. An exemplary case of this is Echinoidea, a clade comprising sea urchins, heart urchins, sand dollars, and allies, for which phylogenomic trees have questioned the timing of previously well-constrained nodes (*Mongiardino Koch et al., 2018*; *Mongiardino Koch and Thompson, 2021d*).

Echinoids are easily recognized by their spine-covered skeletons or tests, composed of numerous tightly interlocking plates. Slightly over 1000 living species have been described to date (*Kroh and Mooi, 2020*), a diversity that populates every marine benthic environment from intertidal to abyssal depths (*Schultz, 2015*). Echinoids are usually subdivided into two morpho-functional groups with similar species-level diversities: 'regular' sea urchins, a paraphyletic assemblage of hemispherical, epibenthic consumers protected by large spines; and irregulars (Irregularia), a clade of predominantly infaunal and bilaterally symmetrical forms covered by small and specialized spines. In today's oceans, regular echinoids act as ecosystem engineers in biodiverse coastal communities such as coral reefs (*Edmunds and Carpenter, 2001*) and kelp forests (*Harrold and Pearse, 1987*), where they are often the main consumers. They are first well known in the fossil record on either side of the Permian-Triassic (P-T) mass extinction event when many species occupied reef environments similar to those inhabited today by their descendants (*Zonneveld et al., 2016*; *Thompson et al., 2017b*). This extinction event was originally thought to have radically impacted the macroevolutionary history of the clade, decimating the echinoid stem group and leading to the radiation of crown group taxa from a single surviving lineage (*Kier, 1977b*; *Twitchett and Oji, 2005*). However, it is now widely accepted that the origin of crown group Echinoidea (i.e., the divergence between its two main lineages, Cidaroidea and Euechinoidea) occurred in the Late Permian, as supported by molecular estimates of divergence (*Smith et al., 2006*; *Thompson et al., 2017a*), as well as the occurrence of Permian fossils with morphologies typical of modern cidaroids (*Smith and Hollingworth, 1990*; *Thompson et al., 2015*). However, a recent total-evidence study recovered many taxa previously classified as crown group members along the echinoid stem, while also suggesting that up to three crown group lineages survived the P-T mass extinction (*Mongiardino Koch and Thompson, 2021d*). This result increases the discrepancy between molecular estimates and the fossil record and renders uncertain the early evolutionary history of crown group echinoids. Constraining the timing of origin of this clade relative to the P-T mass extinction (*Mongiardino Koch et al., 2018*; *Mongiardino Koch and Thompson, 2021d*) is further complicated by the poor preservation potential of stem group echinoids, and the difficulty assigning available disarticulated remains from the Late Paleozoic and Early Triassic to specific clades (*Kier, 1977b*; *Twitchett and Oji, 2005*; *Smith, 2007*; *Kroh and Smith, 2010*; *Thompson et al., 2018*; *Thompson et al., 2019*).

Compared to the morphological conservatism of regular sea urchins, the evolutionary history of the relatively younger Irregularia was characterized by dramatic levels of morphological and ecological innovation (*Kier, 1982*; *Saucède et al., 2006*; *Barras, 2008*; *Hopkins and Smith, 2015*). Within the diversity of irregulars, sand dollars are the most easily recognized (*Figure 1*). The clade includes greatly flattened forms that live in high-energy sandy environments where they feed using a unique mechanism for selecting and transporting organic particles to the mouth, where these are crushed using well-developed jaws (*Mooi, 1990a*; *Nebelsick, 2020*). Sand dollars (Scutelloida) were long thought to be most closely related to sea biscuits (Clypeasteroida) given a wealth of shared morphological characters (*Mooi, 1990a*; *Kroh and Smith, 2010*). The extraordinary fossil record of both sand dollars and sea biscuits suggested their last common ancestor originated in the early Cenozoic from among an assemblage known as 'cassiduloids' (*Mooi, 1990a*; *Saucède et al., 2006*), a once diverse group that is today represented by three depauperate lineages: cassidulids (and close

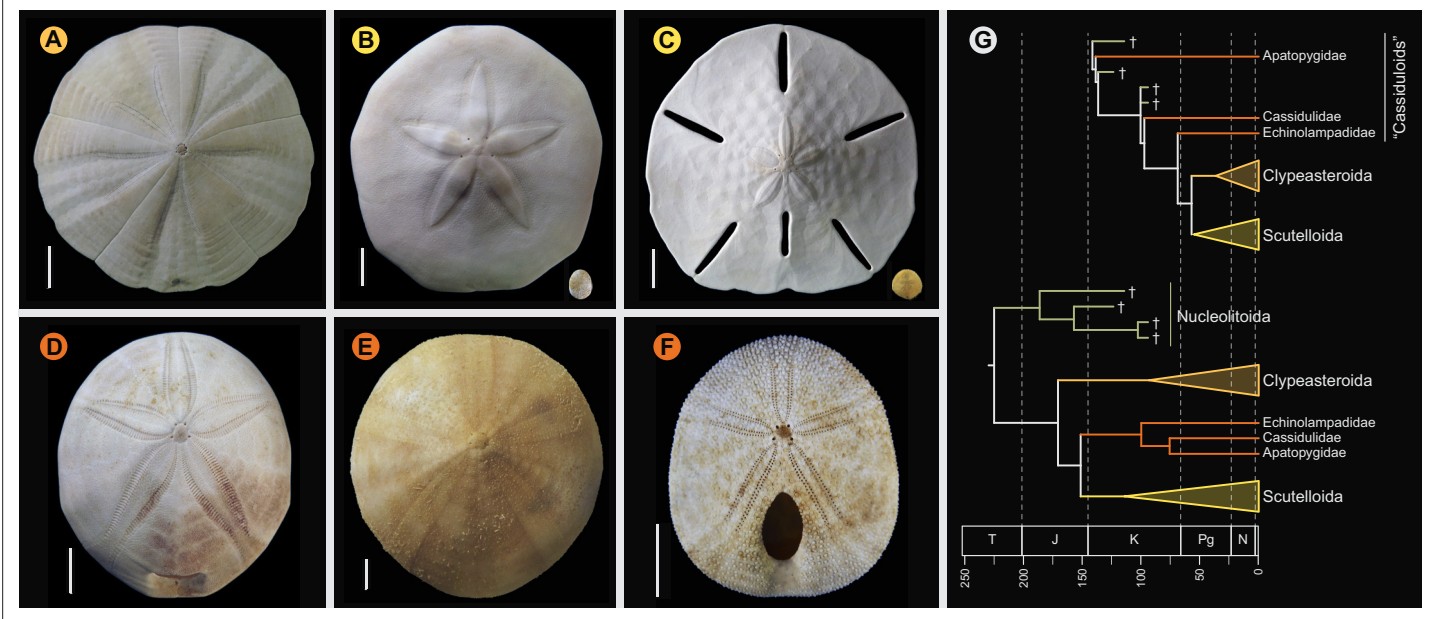

**Figure 1.** Neognathostomate diversity and phylogenetic relationships. (**A**) *Fellaster zelandiae*, North Island, New Zealand (Clypeasteroida).
(**B**) Large specimen: *Peronella japonica*, Ryukyu Islands, Japan; Small specimen: *Echinocyamus crispus*, Maricaban Island, Philippines (Laganina:
Scutelloida). (**C**) Large specimen: *Leodia sexiesperforata*, Long Key, Florida; Small specimen: *Sinaechinocyamus mai*, Taiwan (Scutellina: Scutelloida).
(**D**) *Rhyncholampas pacificus,* Isla Isabela, Galápagos Islands (Cassidulidae). (**E**) *Conolampas sigsbei*, Bimini, Bahamas (Echinolampadidae).
(**F**) *Apatopygus recens*, Australia (Apatopygidae). (**G**) Hypotheses of relationships among neognathostomates. Top: Morphology supports a clade of
Clypeasteroida + Scutelloida originating after the Cretaceous-Paleogene (K-Pg) boundary, subtended by a paraphyletic assemblage of extant (red) and
extinct (green) 'cassiduloids' (***Kroh and Smith, 2010***). Bottom: A recent total-evidence study split cassiduloid diversity into a clade of extant lineages
closely related to scutelloids, and an unrelated clade of extinct forms (Nucleolitoida; ***Mongiardino Koch and Thompson, 2021d***). Divergence times are
much older and conflict with fossil evidence. Cassidulids and apatopygids lacked molecular data in this analysis. Scale bars = 10 mm.

relatives), echinolampadids, and apatopygids (***Smith, 2016***; ***Kroh and Smith, 2010***). These taxa not only lack the defining features of both scutelloids and clypeasteroids but have experienced little morphological change since their origin deep in the Mesozoic (***Kier, 1962***; ***Smith, 2016***; ***Hopkins and Smith, 2015***; ***Souto et al., 2019***). However, early molecular phylogenies supported both cassidulids and echinolampadids as close relatives of sand dollars (e.g., ***Littlewood and Smith, 1995***; ***Smith et al., 2006***), a topology initially disregarded for its conflicts with both morphological and paleontological evidence, but later confirmed using phylogenomic approaches (***Mongiardino Koch et al., 2018***). While many of the traits shared by sand dollars and sea biscuits have since been suggested to represent a mix of convergences and ancestral synapomorphies secondarily lost by some 'cassiduloids' (***Mongiardino Koch et al., 2018***; ***Mongiardino Koch and Thompson, 2021d***), the strong discrepancy between molecular topologies and the fossil record remains unexplained. Central to this discussion is the position of apatopygids, a clade so far unsampled in molecular studies. Apatopygids have a fossil record stretching more than 100 million years and likely have phylogenetic affinities with even older extinct lineages (***Kier, 1962***; ***Kroh and Smith, 2010***; ***Souto et al., 2019***; ***Mongiardino Koch and Thompson, 2021d***). Although current molecular topologies already imply ghost ranges for scutelloids and clypeasteroids that necessarily extend beyond the Cretaceous-Paleogene (K-Pg) boundary, the phylogenetic position of apatopygids could impose even earlier ages on these lineages (***Figure 1***). Constraining these divergences is necessary to understand the timing of origin of the sand dollars, one of the most specialized lineages of echinoids (***Mooi, 1990a***; ***Smith, 2016***; ***Hopkins and Smith, 2015***; ***Nebelsick, 2020***). Resolving some phylogenetic relationships within scutelloids has also been complicated by their recurrent miniaturization and associated loss of morphological features (***Figure 1***; ***Mooi, 1990a***; ***Mooi, 1990b***; ***Mongiardino Koch, 2021a***).

Echinoidea constitutes a model clade in developmental biology and genomics. As these fields embrace a more comparative approach (***Thompson et al., 2017a***; ***Dunn et al., 2018***; ***Smith et al., 2020***), robust and time-calibrated phylogenies are expected to play an increasingly important role.

Likewise, the extraordinary fossil record of echinoids and the ease with which echinoid fossils can be incorporated in phylogenetic analyses make them an ideal system to explore macroevolutionary dynamics using phylogenetic comparative methods (*Mongiardino Koch, 2021a*; *Mongiardino Koch and Thompson, 2021d*). In this study, we build upon available molecular resources with 18 novel genome-scale datasets and build the largest molecular matrix for echinoids yet compiled. Our expanded phylogenomic dataset extends sampling to 16 of the 17 currently recognized echinoid orders – plus the unassigned apatopygids (*Kroh, 2020*) – and is the first to bracket the extant diversity of both sand dollars and sea biscuits and include members of all three lineages of living 'cassiduloids' (cassidulids, echinolampadids, and apatopygids). We also incorporate a diverse sample of outgroups, providing access to the deepest nodes within the crown groups of all other echinoderm classes (holo- thuroids, asteroids, ophiuroids, and crinoids). With it, we reconstruct the phylogenetic relationships and divergence times of the major lineages of living echinoids and place their diversification within the broader context of echinoderm evolution.

## Results

### Phylogeny of Echinoidea

Analyses relied on a 70% occupancy supermatrix composed of 1346 loci (327,695 amino acid sites), and including 54 echinoid terminals plus 12 outgroups. Inference was performed under multiple concatenation and coalescent-aware methodologies, as well as relying on maximum likelihood and Bayesian implementations of site-homogeneous and site-heterogeneous models, as these approaches are known to differ in their susceptibility to model violations (*Lartillot et al., 2007*; *Kainer and Lanfear, 2015*; *Jiang et al., 2020*; see Materials and methods for further details). Phylogenetic relationships supported by the full dataset were remarkably stable, with all nodes but one being identically resolved and fully supported across all methods (*Figure 2A*). While recovering a topology similar to those of previous molecular studies (*Littlewood and Smith, 1995*; *Smith et al., 2006*; *Thompson et al., 2017a*; *Mongiardino Koch et al., 2018*; *Lin et al., 2020*; *Mongiardino Koch and Thompson, 2021d*), this analysis is the first to sample and confidently place micropygoids and aspidodiadematoids within Aulodonta, as well as resolve the relationships among all major clades of Neognathostomata (scutelloids, clypeasteroids and the three lineages of extant 'cassiduloids'). Our results show that *Apatopygus recens* is not related to the remaining 'cassiduloids' but is instead the sister clade to all other sampled neognathostomates. The strong support for this placement, as well as for a clade of cassidulids and echinolampadids (Cassiduloida *sensu stricto*) as the sister group to sand dollars, provides a basis for an otherwise elusive phylogenetic classification of neognathostomates. Our topology also confirms that *Sinaechinocyamus mai*, a miniaturized species once considered a plesiomorphic member of Scutelloida based on the reduction or loss of diagnostic features (*Figure 1*), is in fact a derived paedomorphic lineage closely related to *Scaphechinus mirabilis* (*Mooi, 1990b*).

Salenioida is another major lineage sampled here for the first time, and whose exact position among regular echinoids proved difficult to resolve. While some methods supported salenioids as the sister group to a clade of camarodonts, stomopneustoids, and arbacioids (a topology previously supported by morphology; *Kroh and Smith, 2010*), others recovered a closer relationship of saleni- oids to Camarodonta + Stomopneustoida, with arbacioids sister to them all (as shown in *Figure 2A*). As revealed using likelihood mapping, these results do not stem from a lack of phylogenetic signal, but rather from the presence of strong and conflicting evidence in the dataset regarding the position of salenioids (*Figure 2B*). However, a careful dissection of these signals shows that loci with high phylo- genetic usefulness (as defined by *Mongiardino Koch, 2021b*; *Mongiardino Koch and Thompson, 2021d*; see Materials and methods) favor the topology shown in *Figure 2A*, with the morphological hypothesis becoming dominant only after incorporating less reliable loci (*Figure 2C*). In line with these results, moderate levels of gene subsampling (down to 500 loci) targeting the most phyloge- netically useful loci unambiguously support the placement of arbacioids as sister to the remaining taxa, regardless of the chosen method of inference (*Figure 2D*). More extreme subsampling (down to 100 loci) again results in disagreement among methods. This possibly stems from the increasing effect of stochastic errors in smaller datasets, as less than half of the sampled loci in these reduced datasets contain data for all branches of this quartet (see *Figure 2C*). This result shows the importance of ensuring that datasets (especially subsampled ones) retain appropriate levels of occupancy for

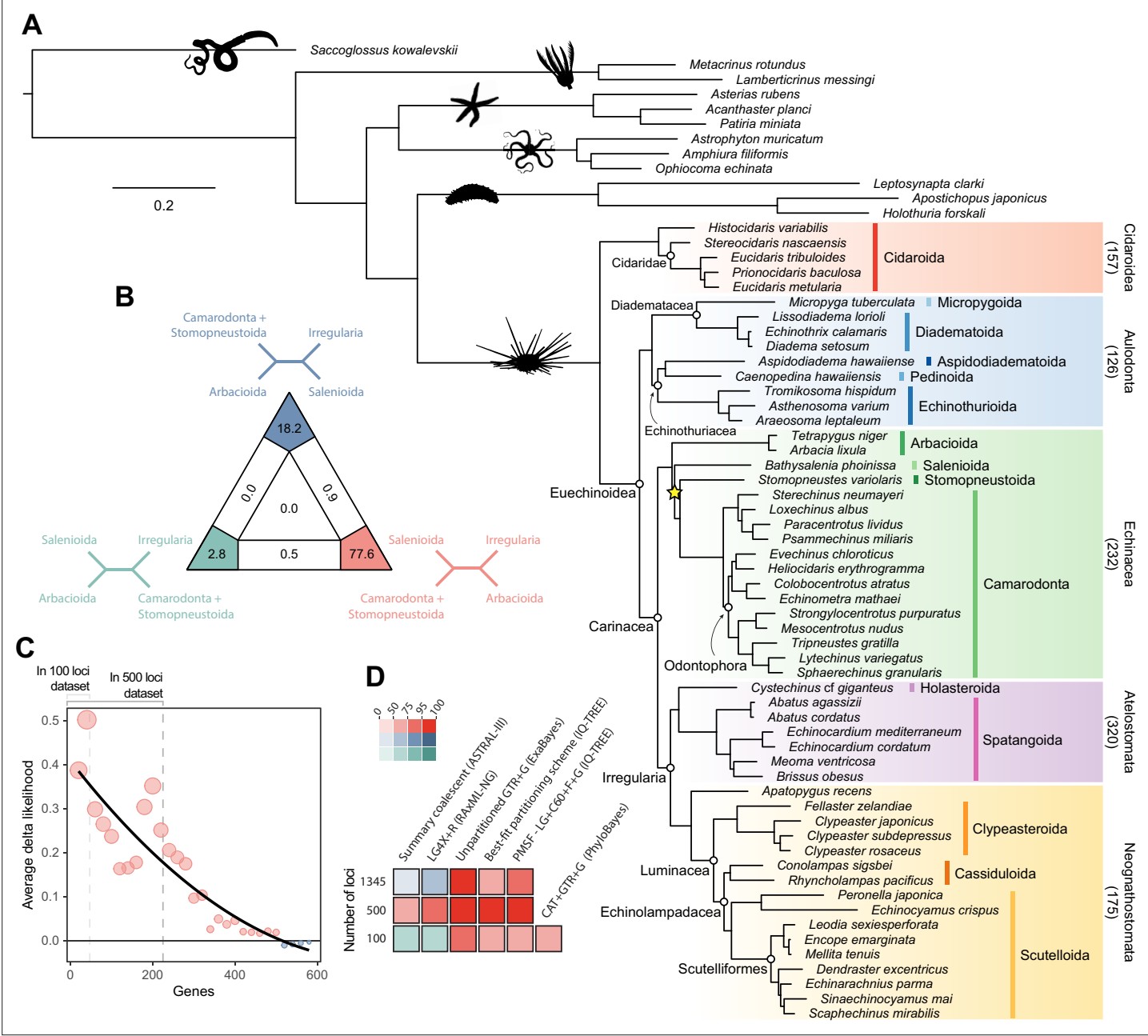

**Figure 2.** Phylogenetic relationships among major clades of Echinoidea. (**A**) Favored topology, as obtained using the full supermatrix and a best-fit partitioning scheme in IQ-TREE (*Nguyen et al., 2015*). With the exception of a single contentious node within Echinacea (marked with a yellow star), all methods supported the same pattern of relationships, and assigned maximum support values to all nodes. Numbers below major clades correspond to the current numbers of described living species (obtained from *Kroh and Mooi, 2020*). (**B**) Likelihood-mapping analysis showing the proportion of quartets supporting different resolutions within Echinacea. While the majority of quartets support the topology depicted in **A** (shown in red), a relatively large number support an alternative resolution that has been recovered in morphological analyses (shown in blue; *Kroh and Smith, 2010*). (**C**) Difference in likelihood score (delta likelihood) for the two resolutions of Echinacea most strongly supported in the likelihood-mapping analysis. Genes were sorted based on their inferred phylogenetic usefulness (*Mongiardino Koch, 2021b*), and gene-wise delta scores were averaged for datasets composed of multiples of 20 loci. Support for a clade of Salenioida + (Camarodonta + Stomopneustoida), as depicted in **A**, is seen as positive delta scores and is predominantly concentrated among the most phylogenetically useful loci. This signal is attenuated in larger datasets that contain less reliable genes, eventually favoring an alternative resolution (as seen by negative scores for the largest datasets). Only the 584 loci containing data for the three main lineages of Echinacea were considered. The line corresponds to a second-degree polynomial regression. (**D**) Resolution and bootstrap scores (see color scale) of the topology within Echinacea found using datasets of different sizes and alternative methods of inference.

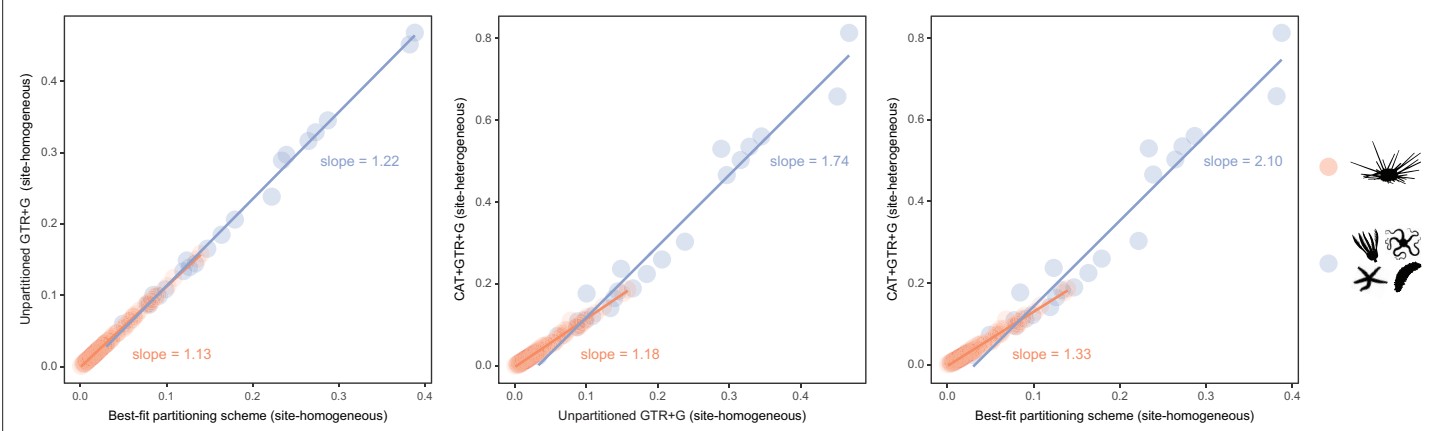

**Figure 3.** Estimated branch lengths across different models of molecular evolution. Different site-homogeneous models (left) infer similar levels of divergence, and the choice between them induces little distortion in the general tree structure. Site-heterogeneous models on the other hand not only infer a larger degree of divergence between terminals relative to site-homogeneous ones (center and right), but they also distort the tree (i.e., impose a non-isometric stretching), with branch lengths connecting outgroup taxa expanding much more than those within the ingroup clade.

clades bracketing contentious nodes (*Dell'Ampio, 2014*). Despite these disagreements, several lines of evidence favor the topology shown in *Figure 2A*, including the results of likelihood mapping, and the increased support for this resolution among the most phylogenetically useful loci and when using more complex methods of reconstruction, such as partitioned and site-heterogeneous models, which always favor this topology regardless of dataset size (*Figure 2D*).

## Sensitivity of node ages

While alternative methods of inference had minor effects on phylogenetic relationships, they did impact the reconstruction of branch lengths (*Figure 3*). Site-heterogeneous models (such as CAT + GTR + G) returned longer branch lengths overall, but also uncovered a larger degree of molecular change among echinoderm classes. Branches connecting these clades were stretched to a much larger extent than those within the ingroup, a phenomenon that might affect the inference of node ages. We tested this hypothesis by exploring the sensitivity of divergence times to the use of alternative models of molecular evolution (site-homogeneous vs. site-heterogeneous), as well as different clocks (autocorrelated vs. uncorrelated), prior node distributions (Cauchy vs. uniform), and gene sampling strategies (using five different approaches; see Materials and methods). All combinations of these factors were explored, resulting in 40 different time calibration settings that were run using Bayesian approaches under a constrained tree topology (shown in *Figure 2A*). While the nodes connecting some outgroup taxa were among those most sensitive to these methodological decisions, large effects were also seen among nodes relating to the origin and diversification of the echinoid clades Cidaroidea, Aulodonta, and Neognathostomata. All of these nodes varied in age by more than 35 Myr – and up to 115 Myr – among the consensus topologies of different analyses (*Figure 4*).

In order to isolate and visualize the impact of each of these factors on divergence time estimation, chronograms were represented in a multidimensional space of node dates, with each axis representing the age of a given node. We term this type of graph a chronospace given its similarities to the treespaces commonly used to explore topological differences among phylogenetic trees (*Hillis et al., 2005*). Each observation (chronogram) was classified as obtained under a specific clock, model of molecular evolution, node prior distribution, and gene sampling strategy, and the major effects of each of these choices were extracted with the use of between-group principal component analyses (bgPCAs). The single dimension of chronospace maximizing the distinctiveness of chronograms obtained under different clocks explained 53.4% of the total variance in node ages across all analyses (*Figure 5*). In contrast, the choice of different loci, models of molecular evolution, and prior distributions on node ages showed much lesser effects, explaining 10.7%, 3.9%, and 0.4% of the total variance, respectively (*Figure 5* and *Figure 5—figure supplement 1*). Even though most of these decisions affected a similar set of sensitive nodes (those mentioned above, as well as some relationships within Atelostomata), the choice of clock model modified the ages of 17 of these by more than

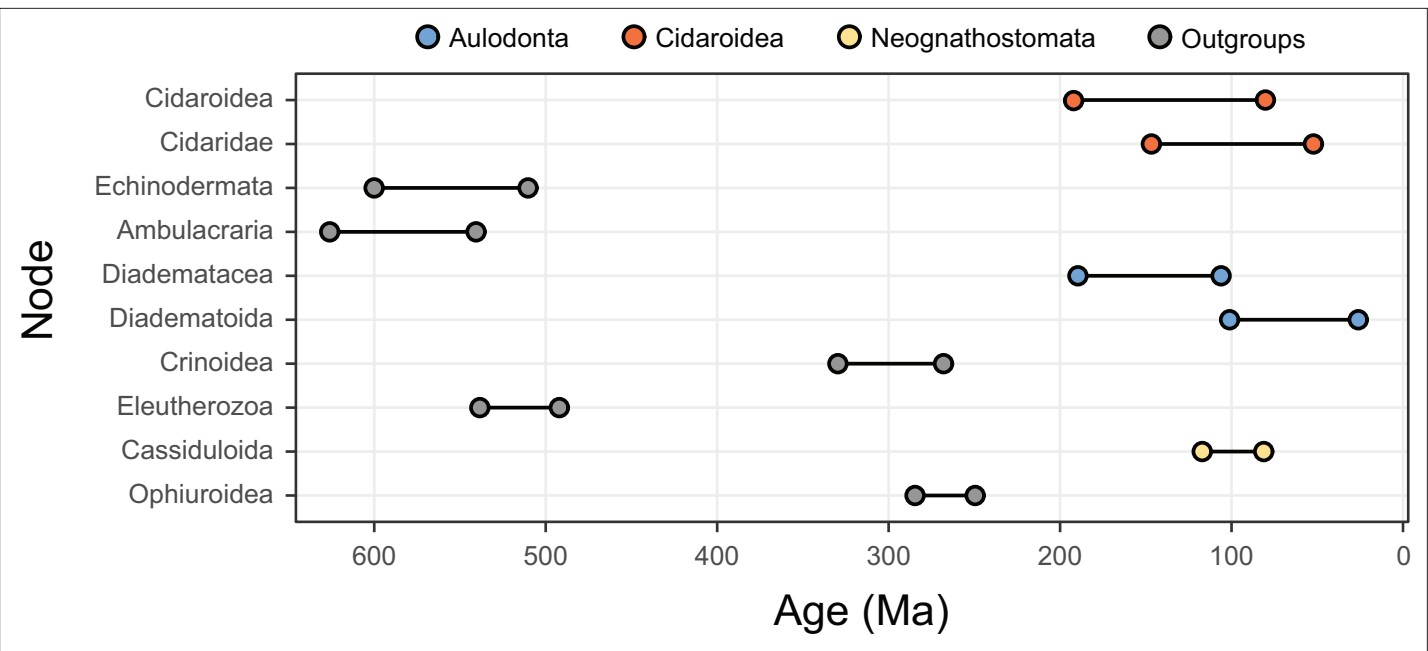

**Figure 4.** The 10 most sensitive node dates are found within Cidaroidea, Aulodonta, Neognathostomata, and among outgroup nodes. For each, the range shown spans the interval between the minimum and maximum ages found among the consensus topologies of the 80 time-calibrated runs performed.

The online version of this article includes the following figure supplement(s) for figure 4:

**Figure supplement 1.** Median ages for selected clades across the consensus trees of the 80 time-calibrated experiments performed.

20 Myr (*Figure 5—figure supplement 2*). This degree of change was induced on only four nodes by selecting alternative loci, and was not induced on any node by enforcing different models of evolution or node age priors (*Figure 5—figure supplements 3–5*). Regarding gene choice, the ages most different to those obtained under random loci selection were found when using the most clock-like genes (*Figure 5C*).

## Echinoid (and echinoderm) divergence times

Even when the age of crown Echinodermata was constrained to postdate the appearance of stereom (the characteristic skeletal microstructure of echinoderms) in the Early Cambrian (*Bottjer et al., 2006*; *Zamora et al., 2013*), only analyses using the most clock-like loci recovered ages concordant with this (i.e., median ages younger than the calibration enforced; *Figure 5—figure supplement 3*). Instead, most consensus trees favored markedly older ages for the clade, in some cases even predating the origin of the Ediacaran biota (*Pu et al., 2016*; *Figure 4—figure supplement 1*). Despite the relative sensitivity of many of the earliest nodes to methodological choices (*Figure 4* and *Figure 4—figure supplement 1*), the split between Crinoidea and all other echinoderms (Eleutherozoa) is always inferred to have predated the end of the Cambrian (youngest median age = 492.1 Ma), and the divergence among the other major lineages (classes) of extant echinoderms are constrained to have happened between the Late Cambrian and Middle Ordovician (*Figure 4—figure supplement 1*). Our results also recover an early origin of crown group Holothuroidea (sea cucumbers; range of median ages = 350.4–384.2 Ma), well before the crown groups of other extant echinoderm classes. These dates markedly postdate the first records of holothuroid calcareous rings in the fossil record (*Reich, 2015*; *Miller et al., 2017*), and imply that this trait does not define the holothuroid crown group but instead evolved from an echinoid-like jaw-apparatus along its stem (*Rahman et al., 2019*). The other noteworthy disagreement between our results and those of previous studies (*Rouse et al., 2013*) involves dating crown group Crinoidea to times that precede the P-T mass extinction (range of median ages = 268.0–329.7 Ma, although highest posterior density intervals are always wide and include Triassic ages).

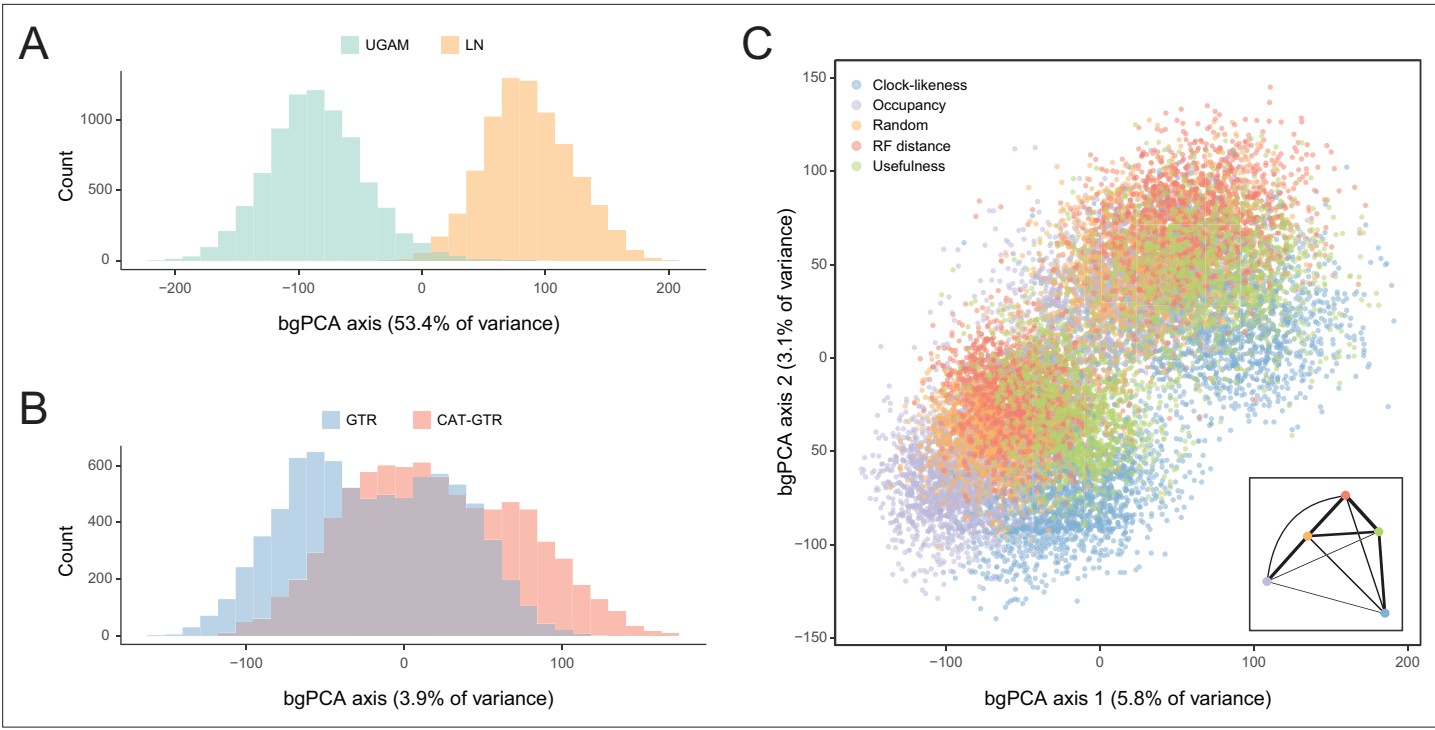

**Figure 5.** Sensitivity of divergence time estimation to methodological decisions. Between-group principal component analysis (bgPCA) was used to retrieve axes that separate chronograms based on the clock model (**A**), model of molecular evolution (**B**), and gene sampling strategy (**C**) employed. In the latter case, only the first two out of four bgPCA dimensions are shown. The inset shows the centroid for each loci sampling strategy, and the width of the lines connecting them are scaled to the inverse of the Euclidean distances that separates them (as a visual summary of overall similarity). The proportions of total variance explained are shown on the axis labels. The impact of the clock model is such that a bimodal distribution of chronograms can be seen even when bgPCA are built to discriminate based on other factors (as in **C**).

The online version of this article includes the following figure supplement(s) for figure 5:

**Figure supplement 1.** Sensitivity of divergence time estimation to the use of alternate prior distributions on calibrated nodes.

**Figure supplement 2.** Distribution of posterior probabilities for node ages that show an average difference larger than 20 Myr depending on the choice of clock prior.

**Figure supplement 3.** Distribution of posterior probabilities for node ages that show a maximum difference larger than 20 Myr depending on the gene sampling strategy.

**Figure supplement 4.** Distribution of posterior probabilities for node ages that are the most affected by the choice of model of molecular evolution.

**Figure supplement 5.** Distribution of posterior probabilities for node ages that are the most affected by the choice of prior distributions on calibrated nodes.

Across all of the analyses performed, the echinoid crown group is found to have originated somewhere between the Pennsylvanian and Cisuralian, with 30.2% posterior probability falling within the late Carboniferous and 69.1% within the early Permian (*Figure 6* and *Figure 6—figure supplement 1*). An origin of the clade postdating the P-T mass extinction is never recovered, even when such ages are common under the joint prior (*Figure 6—figure supplement 2*). While the posterior distribution of ages for Euechinoidea spans both sides of the P-T boundary, the remaining earliest splits within the echinoid tree are constrained to have occurred during the Triassic, including the origins of Aulodonta, Carinacea, Echinacea, and Irregularia (*Figure 6* and *Figure 4—figure supplement 1*). Many echinoid orders are also inferred to have diverged from their respective sister clades during this period, including aspidodiadematoids, pedinoids, echinothurioids, arbacioids, and salenioids. Lineage-through-time plots confirm that diversification proceeded rapidly throughout the Triassic (*Figure 6B*). Despite the topological reorganization of Neognathostomata, the clade is dated to a relatively narrow time interval in the Late to Middle Jurassic (range of median ages = 169.48–180.93 Ma), in agreement with recent estimates (*Mongiardino Koch and Thompson, 2021d*). Within this clade, the origins of both scutelloids and clypeasteroids confidently predate the K-Pg mass extinction (posterior probability of

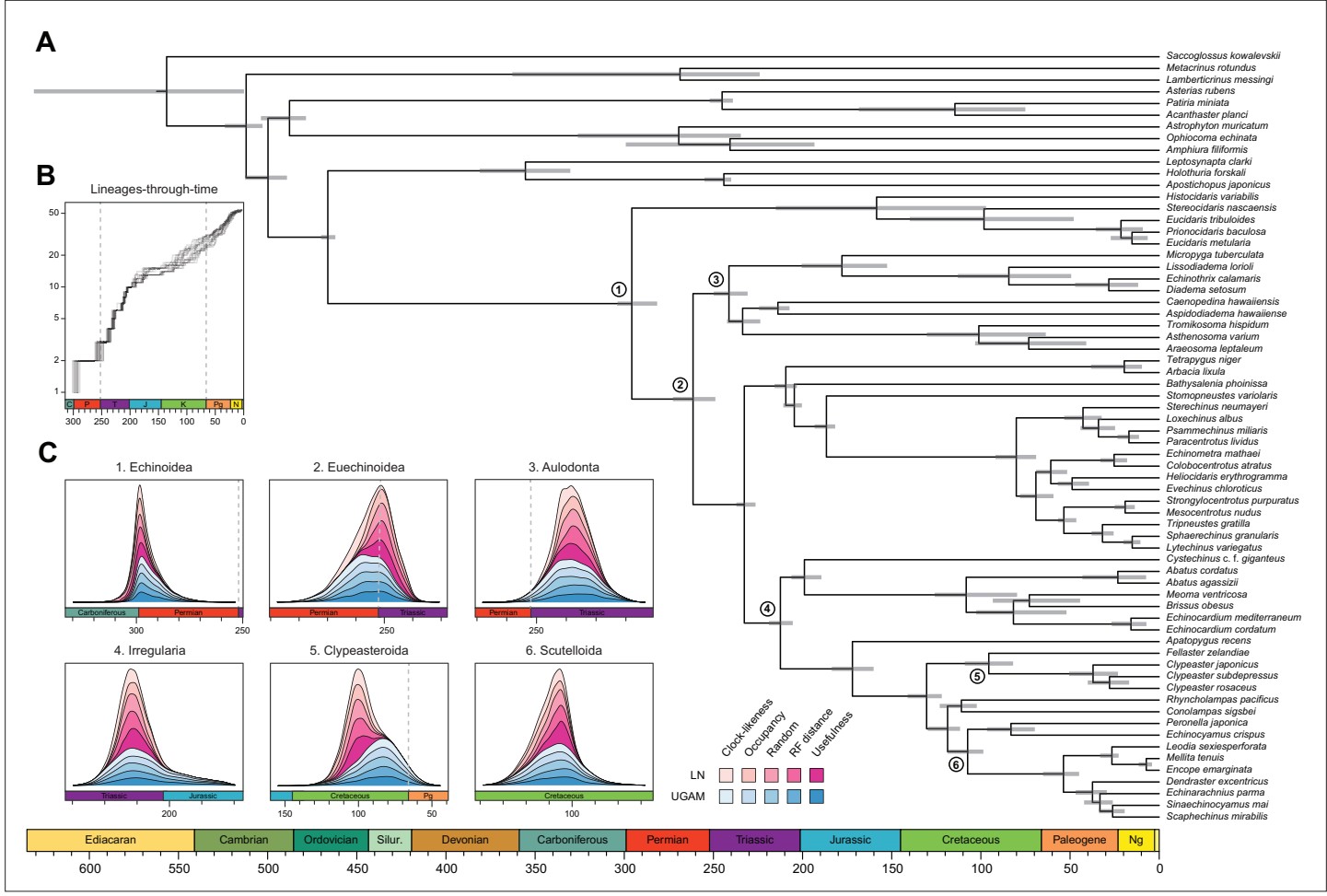

**Figure 6.** Divergence times among major clades of Echinoidea and other echinoderms. (**A**) Consensus chronogram of the two PhyloBayes (*Lartillot et al., 2013*) runs using clock-like genes under a CAT + GTR + G model of evolution, an autocorrelated log-normal (LN) clock, and Cauchy prior distributions. Node ages correspond to median values, and bars show the 95% highest posterior density intervals. (**B**) Lineage-through-time plot, showing the rapid divergence of higher-level clades following the P-T mass extinction (shown with dashed lines, along with the Cretaceous-Paleogene [K-Pg] boundary). Each line corresponds to an individual consensus topology from among the 80 time-calibrated runs performed. (**C**) Posterior distributions of the ages of selected nodes (identified in **A** with numbers). The effects introduced by the use of different models of molecular evolution and node age prior distributions are not shown, as they represent the least important factors (see *Figure 5*); the posterior distributions obtained under different settings of these were merged for every combination of targeted loci and clock prior. Tick marks = 10 Myr.

The online version of this article includes the following figure supplement(s) for figure 6:

**Figure supplement 1.** Number of lineages inferred to have crossed the Permian-Triassic (P-T) boundary.

**Figure supplement 2.** Prior distributions of all constrained nodes.

origination before the boundary = 1.00 and 0.97, respectively), despite younger ages being allowed by the joint prior (*Figure 6—figure supplement 2*).

## Discussion

### The echinoid tree of life

In agreement with previous phylogenomic studies (*Mongiardino Koch et al., 2018*; *Mongiardino Koch and Thompson, 2021d*), echinoid diversity can be subdivided into five major clades (*Figure 2A*). Cidaroids form the sister group to all other crown group echinoids (Euechinoidea). Some aspects of the relationships among sampled cidaroids are consistent with previous molecular (*Brosseau et al., 2012*) and morphological studies (*Kroh and Smith, 2010*), including an initial split between

*Histocidaris* and the remaining taxa, representing the two main branches of extant cidaroids (**Kroh, 2020**; **Kroh and Mooi, 2020**). Others, such as the nested position of *Prionocidaris baculosa* within the genus *Eucidaris,* not only implies paraphyly of this genus but also suggests the need for a taxonomic reorganization of the family Cidaridae. Within euechinoids, the monophyly of Aulodonta is supported for the first time with sampling of all of its major groups. The subdivision of these into a clade that includes diadematoids plus micropygoids (which we propose should retain the name Diadematacea), sister to a clade including echinothurioids and pedinoids (Echinothuriacea *sensu* **Mongiardino Koch et al., 2018**) is strongly reminiscent of some early classifications (e.g., **Durham and Melville, 1957**). Our expanded phylogenomic sampling also confirms an aulodont affinity for aspidodiadematoids (**Kroh, 2020**; **Mongiardino Koch and Thompson, 2021d**) and places them within Echinothuriacea as the sister group to Pedinoida.

The remaining diversity of echinoids, which forms the clade Carinacea (**Figure 2**), is subdivided into Irregularia and their sister clade among regulars, for which we amend the name Echinacea to include Salenioida. Given the striking morphological gap separating regular and irregular echinoids, the origin of Irregularia has been shrouded in mystery (**Durham and Melville, 1957**; **Saucède et al., 2006**; **Kroh and Smith, 2010**). Our complete sampling of major regular lineages determines Echinacea *sensu stricto* to be the sister clade to irregular echinoids. A monophyletic Echinacea was also supported in a recent total-evidence analysis (**Mongiardino Koch and Thompson, 2021d**), but the incomplete molecular sampling of that study resulted in a slightly different topology that placed salenioids as the sister group to the remaining lineages. However, an overall lack of morphological synapomorphies uniting these clades had previously been acknowledged (**Kroh and Smith, 2010**). While the relationships within Echinacea proved to be difficult to resolve even with thousands of loci, multiple lines of evidence lead us to prefer a topology in which salenioids form a clade with camarodonts + stomopneustoids, with arbacioids sister to all of these (**Figure 2**).

As has been already established (**Littlewood and Smith, 1995**; **Smith et al., 2006**; **Kroh and Smith, 2010**; **Mongiardino Koch et al., 2018**; **Mongiardino Koch and Thompson, 2021d**), the lineages of irregular echinoids here sampled are subdivided into Atelostomata (heart urchins and allies) and Neognathostomata (sand dollars, sea biscuits, and 'cassiduloids'). Despite the former being the most diverse of the five main clades of echinoids (**Figure 2**), its representation in phylogenomic studies remains low, and its internal phylogeny poorly constrained (**Kroh, 2020**). On the contrary, recent molecular studies have greatly improved our understanding of the relationships among neognathostomates (**Mongiardino Koch et al., 2018**; **Lin et al., 2020**; **Mongiardino Koch and Thompson, 2021d**), revealing an evolutionary history that dramatically departs from previous conceptions. Even when scutelloids and clypeasteroids were never recovered as reciprocal sister lineages by molecular phylogenies (e.g., **Littlewood and Smith, 1995**; **Smith et al., 2006**; **Thompson et al., 2017a**), this result was not fully accepted until phylogenomic data confidently placed echinolampadids as the sister lineage to sand dollars (**Mongiardino Koch et al., 2018**). At the same time, this result rendered the position of the remaining 'cassiduloids', a taxonomic wastebasket with an already complicated history of classification (**Suter, 1994**; **Smith, 2016**; **Kroh and Smith, 2010**; **Souto et al., 2019**), entirely uncertain. An attempt to constrain the position of these using a total-evidence approach (**Mongiardino Koch and Thompson, 2021d**) subdivided the 'cassiduloids' into three unrelated clades: Nucleolitoida, composed of extinct lineages and placed outside the node defined by Scutelloida + Clypeasteroida, and two other clades nested within it (see **Figure 1G**). Extant 'cassiduloids' were recovered as members of one of the latter clades, representing the monophyletic sister group to sand dollars. Here, we show that *Apatopygus recens* does not belong within this clade but is instead the sister group to all other extant neognathostomates. Given this phylogenetic position, as well as the morphological similarities between *Apatopygus* and the entirely extinct nucleolitids (**Mortensen, 1948**; **Kier, 1966**; **Suter, 1994**; **Kroh and Smith, 2010**; **Souto et al., 2019**), it is likely that the three extant species of apatopygids represent the last surviving remnants of Nucleolitoida, a clade of otherwise predominantly Mesozoic neognathostomates (**Mongiardino Koch and Thompson, 2021d**). Because of the renewed importance in recognizing this topology, we propose the name Luminacea for the clade uniting all extant neognathostomates with the exclusion of Apatopygidae (**Figure 2A**). This nomenclature refers to the dynamic evolutionary history of the Aristotle's lantern (i.e., the echinoid jaw-apparatus) within the clade (present in the adults of both clypeasteroids and scutelloids, but found only in the juveniles of Cassiduloida *sensu stricto*), the inclusion of the so-called lamp urchins (echinolampadids) within the

clade, and the illumination provided by this hitherto unexpected topology. The previous misplacement of *Apatopygus* (*Mongiardino Koch and Thompson, 2021d*; see *Figure 1G*) is likely a consequence of tip-dating preferring more stratigraphically congruent topologies (*King, 2020*), an effect that can incorrectly resolve taxa on long terminal branches (*Turner et al., 2017*). Given the generally useful phylogenetic signal of stratigraphic information (*Mongiardino Koch et al., 2021c*), this inaccuracy further highlights the unusual evolutionary history of living apatopygids.

## Chronospaces: a statistical exploration of time calibration strategies

Calibrating phylogenies to absolute time is crucial to understanding evolutionary history, as the resulting chronograms provide a major avenue for testing hypotheses of diversification, character evolution, and other macroevolutionary processes. However, the accuracy and precision of the inferred divergence times hinge upon many methodological choices (calibration strategies, prior distributions on node ages, clock models, etc.), that are often difficult or time-consuming to justify (*Warnock et al., 2012*; *Sauquet, 2013*; *dos Reis et al., 2015*; *Reis et al., 2018*; *Carruthers et al., 2020*; *Carruthers and Scotland, 2021*), and whose impact can be hard to quantify.

Here, we analyze the sensitivity of node ages to alternative criteria to sample loci from phylogenomic datasets, as well as different assumptions regarding patterns of molecular evolution across sites, variation in evolutionary rates among lineages, and ways in which fossils are translated into plausible times of divergence. To do so, we introduce an approach to visualize the distribution of chronograms in a multidimensional space of node ages, a chronospace, and measure the overall effect of these decisions on inferred dates using multivariate statistical methods. Our results reveal a minimal impact of selecting between alternative distributions to model the prior ages of calibrated nodes. This result conflicts with previous results (e.g., *Inoue et al., 2010*; *dos Reis et al., 2015*; *Strassert et al., 2021*), and may reflect the way these distributions are implemented in the software employed (Phylo-Bayes v4.1; *Lartillot et al., 2013*). Similarly, divergence times obtained under site-homogeneous and site-heterogeneous models (such as CAT + GTR + G) are broadly comparable. This happens despite the latter estimating higher levels of sequence divergence and stretching branches in a non-isometric manner (*Figure 3*). While site-heterogeneous models have become common for the inference of phylogenetic relationships, the degree to which they impact estimates of node ages has received less scrutiny. The lack of a meaningful effect uncovered here, coupled with their high computational burden (*Whelan and Halanych, 2017*), questions their usefulness for time-scaling phylogenies. A similar result was recently found when comparing site-homogeneous models with different numbers of parameters (*Tao et al., 2020*), suggesting that relaxed clocks adjust branch rates in a manner that buffers the effects introduced by using more complex models of sequence evolution.

The choice between different loci also has a small effect on inferred ages, with little evidence of a systematic difference between the divergence times supported by randomly chosen loci and those found using targeted sampling criteria, such as selecting genes for their phylogenetic signal, usefulness, occupancy, or clock-likeness. A meaningful effect was restricted to a few ancient nodes (e.g., Echinodermata), for which clock-like genes suggested younger ages that are more consistent with fossil evidence. While this validates the use of clock-like genes for inferring deep histories of diversification (*Smith et al., 2018*; *Carruthers et al., 2020*), the choice of loci had no meaningful effect on younger ages. Finally, the choice between alternative clock models induced differences in ages that were between five and one hundred times stronger than those of other factors, emphasizing the importance of either validating their choice (e.g., *Tao et al., 2019*) or – as done here – focusing on results that are robust to them.

## Echinoid origins and diversification

The origin and early diversification of crown group Echinoidea have always been considered to have been determined (or at least strongly affected) by the P-T mass extinction (*Kier, 1977b*; *Twitchett and Oji, 2005*; *Thompson et al., 2015*; *Thompson et al., 2019*). However, estimating the number of crown group members surviving the most severe biodiversity crisis in the Phanerozoic (*Raup and Sepkoski, 1982*) has been hampered by both paleontological and phylogenetic uncertainties (*Smith and Hollingworth, 1990*; *Smith et al., 2006*; *Thompson et al., 2017a*; *Thompson et al., 2017b*; *Thompson et al., 2018*; *Mongiardino Koch and Thompson, 2021d*). Our results establish that multiple crown group lineages survived and crossed this boundary, finding for the first time a null

posterior probability of the clade originating after the extinction event. While the survival of three crown group lineages is slightly favored (*Figure 6—figure supplement 1*), discerning between alternative scenarios is still precluded by uncertainties in dating these early divergences. Echinoid diversification during the Triassic was relatively fast (*Figure 6B*) and involved rapid divergences among its major clades. Even many lineages presently classified at the ordinal level trace their origins to this initial pulse of diversification following the P-T mass extinction.

The late Paleozoic and Triassic origins inferred for the crown group and many euechinoid orders prompt a re-evaluation of fossils from this interval of time. Incompletely known fossil taxa such as the Pennsylvanian *Eotiaris? meurevillensis*, with an overall morphology akin to that of crown group echinoids, has a stratigraphic range consistent with our inferred date for the origin of the echinoid crown group (*Thompson et al., 2019*). Additionally, the Triassic fossil record of echinoids has been considered to be dominated by stem group cidaroids, with the first euechinoids not known until the Late Triassic (*Kier, 1984*; *Smith, 2007*). However, the Triassic origins of many euechinoid lineages supported by our analyses necessitate that potential euechinoid affinities should be re-considered for this diversity of Triassic fossils. This is especially the case for the serpianotiarids and triadocidarids, abundant Triassic families variously interpreted as cidaroids, euechinoids, or even stem echinoids (*Kier, 1984*; *Smith, 1994*; *Mongiardino Koch and Thompson, 2021d*). A reinterpretation of any of these as euechinoids would suggest that the long-implied gap in the euechinoid record (*Thompson et al., 2015*; *Thompson et al., 2018*) is caused by our inability to correctly place these key fossils, as opposed to an incompleteness of the fossil record itself.

While our phylogenomic approach is the first to resolve the position of all major cassiduloid lineages, the inferred ages for many nodes within Neognathostomata remain in strong disagreement with the fossil record. No Mesozoic fossil can be unambiguously assigned to either sand dollars or sea biscuits, a surprising situation given the good fossilization potential and highly distinctive morphology of these clades (*Kier, 1977a*, *Kier, 1982*; *Mooi, 1990a*; *Kroh and Smith, 2010*). While molecular support for a sister group relationship between scutelloids and echinolampadids already implied this clade (Echinolampadacea) must have split from clypeasteroids by the Late Cretaceous (*Smith et al., 2006*; *Kroh and Smith, 2010*; *Mongiardino Koch et al., 2018*; *Mongiardino Koch and Thompson, 2021d*), this still left open the possibility that the crown groups of sand dollars and sea biscuits radiated in the Cenozoic. Under this scenario, the Mesozoic history of these groups could have been composed of stem forms lacking their distinctive morphological features, complicating their correct identification. This hypothesis is here rejected, with the data unambiguously supporting the origination of the sand dollar and sea biscuit crown groups preceding the K-Pg mass extinction (*Figure 6C*). While it remains possible that these results are incorrect even after such a thorough exploration of the time calibration toolkit (see for example *Carruthers et al., 2020*; *Field et al., 2020*), these findings call for a critical reassessment of the Cretaceous fossil record, and a better understanding of the timing and pattern of morphological evolution among fossil and extant neognathostomates. For example, isolated teeth with an overall resemblance to those of modern sand dollars and sea biscuits have been found in Lower Cretaceous deposits (*Caramés et al., 2019*), raising the possibility that other overlooked and disarticulated remains might close the gap between rocks and clocks.

## Conclusions

Although echinoid phylogenetics has long been studied using morphological data, the position of several major lineages (e.g., aspidodiadematoids, micropygoids, salenioids, apatopygids) remained to be confirmed with the use of phylogenomic approaches. Our work not only greatly expands the available genomic resources for the clade, but finds novel resolutions for some of these lineages, improving our understanding of their evolutionary history. The most salient aspect of our topology is the splitting of the extant 'cassiduloids' into two distantly related clades, one of which is composed exclusively of apatopygids. This result is crucial to constrain the ancestral traits shared by the main lineages of neognathostomates, helping unravel the evolutionary processes that gave rise to the unique morphology of the sand dollars and sea biscuits (*Mooi, 1990a*; *Hopkins and Smith, 2015*; *Mongiardino Koch and Thompson, 2021d*).

Although divergence time estimation is known to be sensitive to many methodological decisions, systematically quantifying the relative impact of these on inferred ages has rarely been done. Here, we propose an approach based on chronospaces that can help visualize key effects and determine

the sensitivity of node dates to different assumptions. Our results shed new light on the early evolutionary history of crown group echinoids and its relationship with the P-T mass extinction event, a point in time where the fossil record provides ambiguous answers. They also establish with confidence a Cretaceous origin for the sand dollars and sea biscuits, preceding their first appearance in the fossil record by at least 40–50 Myr, respectively (and potentially up to 65 Myr). These clades, therefore, join several well-established cases of discrepancies between the fossil record and molecular clocks, such as those underlying the origins of placental mammals (*Ronquist et al., 2016*) and flowering plants (*Coiro et al., 2019*).

## Materials and methods
### Sampling, bioinformatics, and matrix construction

This study builds upon previous phylogenomic matrices (*Mongiardino Koch et al., 2018*; *Mongiardino Koch and Thompson, 2021d*), the last of which was augmented through the addition of eight published datasets (mostly expanding outgroup sampling), as well as 18 novel echinoid datasets (16 transcriptomes and 2 draft genomes). For all novel datasets, tissue sampling, DNA/RNA extraction, library preparation, and sequencing varied by specimen, and are detailed in Appendix 1. Raw reads have been deposited in NCBI under Bioproject accession numbers PRJNA767441, PRJNA746411, and PRJNA746412. Final taxon sampling included 12 outgroups and 54 echinoids (SRA accession numbers and sampling details can be found in *Appendix 1—table 1*).

Raw reads for all transcriptomic datasets were trimmed or excluded using quality scores with Trimmomatic v. 0.36 (*Bolger et al., 2014*) under default parameters. Further sanitation steps were performed using the Agalma 2.0 phylogenomic workflow (*Dunn et al., 2013*), and datasets were assembled *de novo* with Trinity v. 2.5.1 (*Grabherr et al., 2011*). For genomic shotgun sequences, adapters were removed with BBDuk (https://sourceforge.net/projects/bbmap/), and UrQt v. 1.0.18 (*Modolo and Lerat, 2015*) was used to filter short reads (size <50) and trim low-quality ends (score <28). Datasets were then assembled using MEGAHIT v. 1.1.2 (*Li et al., 2015*). Draft genomes were masked using RepeatMasker v. 4.1.0 (*Smit et al., 2015*; *Hoff and Stanke, 2019*), before obtaining gene predictions with AUGUSTUS v. 3.2.3 (*Stanke et al., 2006*). A custom set of universal single-copy orthologs (USCOs) obtained from the *Strongylocentrotus purpuratus* genome assembly v. 5.0 was employed as the training dataset. Settings and further details of these analyses can be found in Appendix 1.

Multiplexed transcriptomes were sanitized from cross-contaminants using CroCo v. 1.1 (*Simion et al., 2018*), and likely non-metazoan contaminants were removed using alien_index v. 3.0 (*Ryan, 2014*), removing sequences with AI scores > 45. Datasets were imported back into Agalma, which automated orthology inference (as described in *Dunn et al., 2013*; *Guang et al., 2021*), gene alignment with MAFFT v. 7.305 (*Katoh and Standley, 2013*; using the E-INS-i algorithm), and trimming with GBLOCKS v. 0.91b (*Talavera and Castresana, 2007*). The amino acid supermatrix was reduced using a 70% occupancy threshold, producing a final dataset of 1346 loci (327,695 sites). As a final sanitation step, gene trees were obtained using ParGenes v. 1.0.1 (*Morel et al., 2018*), which performed model selection (minimizing the Bayesian information criterion) and phylogenetic inference using 100 bootstrap (BS) replicates. Trees were then used to remove outlier sequences with TreeShrink v. 1.3.1 (*Mai and Mirarab, 2018*). We specified a reduced tolerance for false positives and limited removal to at most three terminals which had to increase tree diameter by at least 25% (-q 0.01 -k 3 -b 25). Statistics for the supermatrix and all assemblies can be found in *Appendix 1—table 2*.

### Phylogenetic inference

Coalescent-based inference was performed using the summary method ASTRAL-III (*Zhang et al., 2018*), estimating support as local posterior probabilities (*Sayyari and Mirarab, 2016*). Among concatenation approaches, we used Bayesian inference under an unpartitioned GTR + G model in ExaBayes v. 1.5 (*Aberer et al., 2014*). Two chains were run for 2.5 million generations, samples were drawn every one hundred, and the initial 25% was discarded as burn-in. The entire posterior distribution of trees was composed of a single topology, and 210 out of 213 parameters attained adequate potential scale reduction factors (i.e., lower than 1.1). We also explored maximum likelihood inference with partitioned and unpartitioned models. For the former, the fast-relaxed clustering algorithm was used to find the best-fitting model among the top 10% using IQ-TREE v. 1.6.12 (*Nguyen et al., 2015*;

*Kalyaanamoorthy et al., 2017*; -m MFP + MERGE -rclusterf 10 -rcluster-max 3000), and support was evaluated with 1000 ultrafast BS replicates (*Hoang et al., 2017*). For the latter, we used the LG4X + R model in RAxML-NG v. 0.5.1 (*Kozlov et al., 2019*) and evaluated support with 200 replicates of BS. Finally, we also implemented the site-heterogeneous LG + C60 + F + G mixture model using the posterior mean site frequency approach to provide a fast approximation of the full profile mixture model (*Wang et al., 2018*), allowing the use of 100 BS replicates to estimate support. Given some degree of topological conflict between the results of the other methods (see below), multiple guide trees were used to estimate site frequency profiles, but the resulting phylogenies were identical.

Given conflicts between methods in the resolution of one particular node (involving the relationships among Arbacioida, Salenioida, and the clade of Stomopneustoida + Camarodonta), all methods were repeated after reducing the matrix to 500 and 100 loci selected for their phylogenetic usefulness using the approach described in *Mongiardino Koch, 2021b*; *Mongiardino Koch and Thompson, 2021d*, and implemented in the *genesortR* script (https://github.com/mongiardino/genesortR). This approach relies on seven gene properties routinely used for phylogenomic subsampling, including multiple proxies for phylogenetic signal – such as the average BS and Robinson-Foulds (RF) similarity to a target topology – as well as several potential sources of systematic bias (e.g., rate and compositional heterogeneity). Outgroups were removed before calculating these metrics. RF similarity was measured to a species tree that had the conflicting relationship collapsed so as not to bias gene selection in favor of any resolution. A PCA of this dataset resulted in a dimension (PC 2, 17.6% of variance) along which phylogenetic signal increased while sources of bias decreased (*Appendix 3—figure 1*), and which was used for loci selection. For the smallest subsampled dataset (100 loci), we also performed inference under the site-heterogeneous CAT + GTR + G model using PhyloBayes-MPI (*Lartillot et al., 2013*). Three runs were continued for >10,000 generations, sampling every two generations and discarding the initial 25%. Convergence was confirmed given a maximum bipartition discrepancy of 0.067 and effective sample sizes for all parameters > 150.

Two other approaches were used in order to assist in resolving the contentious node. First, we implemented a likelihood-mapping analysis (*Strimmer and Von Haeseler, 1997*) in IQ-TREE to visualize the phylogenetic signal for alternative resolutions of the quartet involving these three lineages (Arbacioida, Salenioida, and Stomopneustoida + Camarodonta) and their sister clade (Irregularia; other taxa were excluded). Second, we estimated the log-likelihood scores of each site in RAxML (using best-fitting models) for the two most strongly supported resolutions found through likelihood mapping. These were used to calculate gene-wise differences in scores, or δ values (*Shen et al., 2017*). In order to search for discernable trends in the signal for alternative topologies, genes were ordered based on their phylogenetic usefulness (see above) and the mean per-locus δ values of datasets composed of multiples of 20 loci (i.e., the most useful 20, 40, etc.) were calculated.

## Time calibration

Node dating was performed using relaxed molecular clocks in PhyloBayes v4.1 using a fixed topology and a novel set of 22 fossil calibrations corresponding to nodes from our newly inferred phylogeny (listed in Appendix 2). Depending on the node, we enforced both minimum and maximum bounds, or either one of these. A birth-death prior was used for divergence times, which allowed for the implementation of soft bounds (*Yang and Rannala, 2006*), leaving 5% prior probability of divergences falling outside of the specified interval. We explored the sensitivity of divergence times to gene selection, model of molecular evolution, type of clock, and prior distribution on node ages. One hundred loci were sampled from the full supermatrix according to four targeted sampling schemes: usefulness (calculated as explained above, except incorporating all echinoderm terminals), phylogenetic signal (i.e., smallest RF distance to species tree), clock-likeness (i.e., smallest variance of root-to-tip distances), and level of occupancy. For clock-likeness, we only considered loci that lay within one standard deviation of the mean rate (estimated by dividing total tree length by the number of terminals; *Telford et al., 2014*), as this method is otherwise prone to selecting largely uninformative loci (*Figure 7*; *Mongiardino Koch, 2021b*). A fifth sample of randomly chosen loci was also evaluated. These five datasets were run under two unpartitioned models of molecular evolution, the site-homogeneous GTR + G and the site-heterogeneous CAT + GTR + G, and both uncorrelated gamma (UGAM; *Drummond et al., 2006*) and autocorrelated log-normal (LN; *Thorne et al., 1998*; *Kishino et al., 2001*) clocks were implemented. Finally, fossil calibrations were translated into node age priors

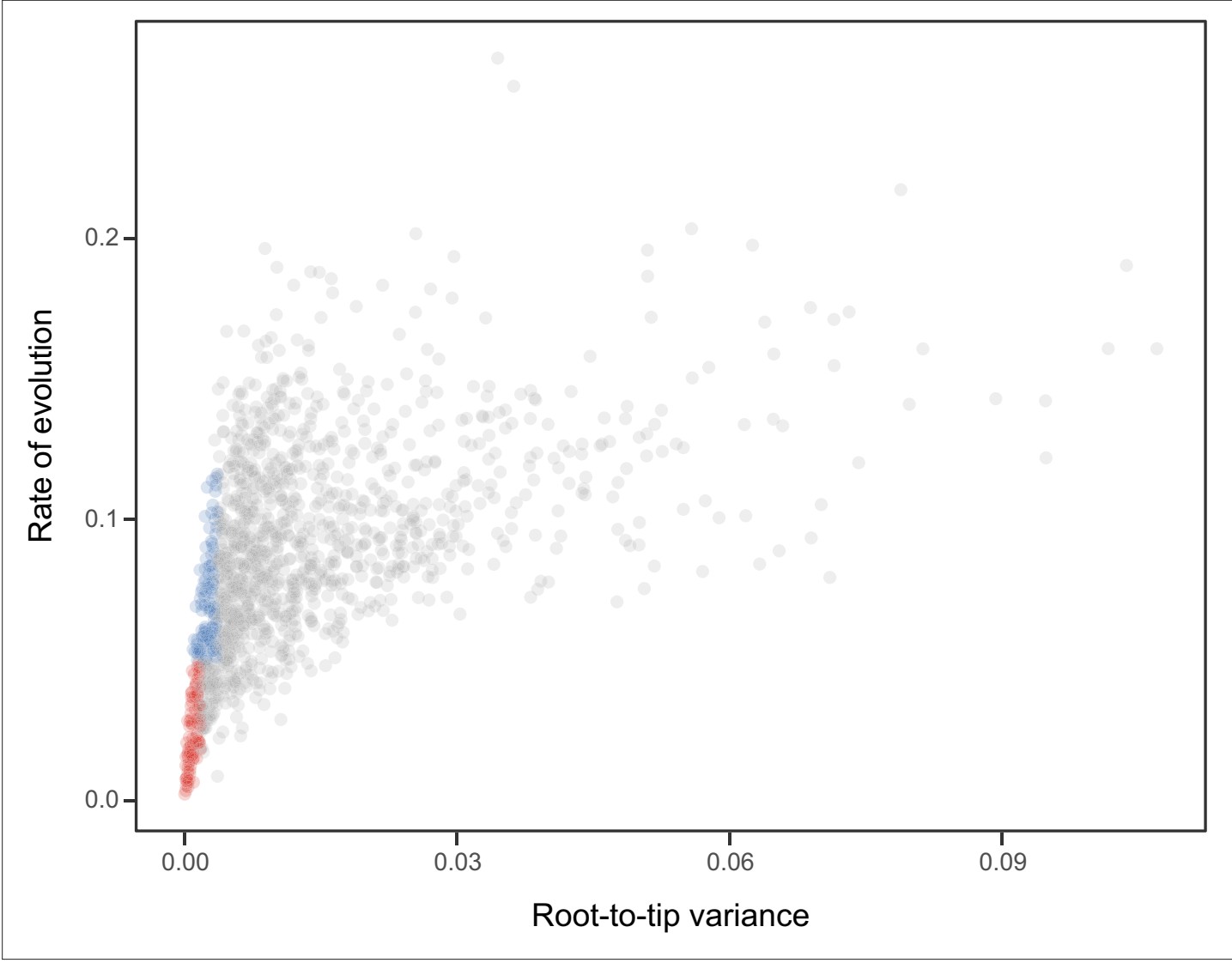

**Figure 7.** Relationship between the root-to-tip variance (a proxy for the clock-likeness of loci) and the rate of evolution. The most clock-like loci (shown in red), which are often favored for the inference of divergence times (e.g., *Smith et al., 2018*; *Carruthers et al., 2020*), are among the most highly conserved and can provide little information for constraining node ages (see also *Mongiardino Koch, 2021b*). Clock-like genes with a higher information content were used instead by choosing the loci with the lowest root-to-tip variance from among those that were within one standard deviation from the mean evolutionary rate (shown in blue).

with the use of both uniform and Cauchy distributions (under default parameters), the latter of which account for the incompleteness of the fossil record by assuming that the most likely origination times occur at a distance from the minimum bound (*dos Reis et al., 2015*). While some methods have been developed to guide the selection of these parameters, exploring the sensitivity of results to a broad spectrum of conditions (even if some are suboptimal) can provide a better picture of the robustness of results to underlying assumptions. Furthermore, guidance methods can also be subject to their own assumptions. For example, CorrTest (*Tao et al., 2019*), an approach to select between alternative clock models, either supported or rejected the presence of autocorrelated rates depending on the species tree used from among those obtained under different methods of phylogenetic inference (see above).

The combination of these settings (loci sampled, model of evolution, type of clock, and node prior distribution) resulted in 40 analyses. For each, two runs were continued for 20,000 generations, after which the initial 25% was discarded and the chains thinned to every two generations (see log-likelihood trace plots in *Appendix 3—figure 2*). To explore the sensitivity of divergence times to these

methodological decisions, 400 random chronograms were sampled from each analysis (200 from each run), and their node dates were subjected to bgPCA using package *Morpho* (*Schlager, 2017*) in the R statistical environment (*R Development Core Team, 2019*). bgPCA involves the use of PCA on the covariance matrix of group means, followed by the projection of original samples onto the obtained axes. The result is a multidimensional representation of divergence times – a chronospace – rotated so as to capture the distinctiveness of observations obtained under different settings. Separate bgPCAs were performed for each of the four factors explored, and the proportion of total variance explained by bgPC axes was taken as an estimate of the relative impact of these choices on divergence times. Finally, lineage-through-time plots were generated using *ape* (*Paradis and Schliep, 2018*).

## Acknowledgements

We are grateful to the crew of R/Vs Antéa, Atlantis, Falkor and Mirai, HOV Alvin, and the crew and PIs of BIOMAGLO deep-sea cruises (Laure Corbari, Karine Olu-Le Roy, Sarah Samadi) for assistance in specimen collection. We would also like to thank Libby Liggins, Wilma Blom, Owen Anderson, Francisco Solís-Marín, Carlos A Conejeros-Vargas, and Jih-Pai Lin for the collection and shipment of specimens. We gratefully acknowledge assistance from Thomas Saucède (Univ. Burgundy), Marc Eléaume (MNHN), and Charlotte Seid (Scripps) in accessing, cataloging, and processing museum specimens. Tim Ravasi provided resources and collection facilities to GWR via King Abdullah University of Science and Technology (KAUST). Institutional support was provided by the Central Research Laboratories and the Department of Geology and Palaeontology at the Natural History Museum Vienna, Austria. We also thank Pablo Milla Carmona for providing statistical advice. This work was supported by a Yale Institute for Biospheric Studies Doctoral Dissertation Improvement Grant and a Society of Systematic Biologists Graduate Student Research Award to NMK. AK received funding from an Austrian Science Fund project (FWF, project number P 29508-B25), FA from Agencia Nacional de Investigación (ANID/ PAI Inserción en la Academia, project number PAI79170033), and GWR and RM from NSF grants DEB-2036186 and DEB-2036298, respectively. JRT was supported by a Royal Society Newton International Fellowship and a Leverhulme Trust Early Career Fellowship. NMK was supported by a Yale University fellowship and NSF grant DEB-2036186.

## Additional information

### Funding

| Funder | Grant reference number | Author |
|---|---|---|
| Yale Institute for Biospheric Studies | Doctoral Dissertation Improvement Grant | Nicolás Mongiardino Koch |
| Society of Systematic Biologists | Graduate Student Research Award | Nicolás Mongiardino Koch |
| Austrian Science Fund | P 29708-B25 | Andreas Kroh |
| Agencia Nacional de Investigación | PAI79170033 | Felipe Aguilera |
| National Science Foundation | DEB-2036186 | Greg W Rouse |
| National Science Foundation | DEB-2036298 | Rich Mooi |
| Agencia Nacional de Investigación y Desarrollo | ANID FONDECYT Iniciación 11180084 | Felipe Aguilera |

The funders had no role in study design, data collection and interpretation, or the decision to submit the work for publication.

### Author contributions

Nicolás Mongiardino Koch, Conceptualization, Data curation, Formal analysis, Funding acquisition, Investigation, Methodology, Project administration, Resources, Software, Validation, Visualization,

Writing – original draft, Writing – review and editing; Jeffrey R Thompson, Data curation, Writing – original draft, Writing – review and editing; Avery S Hiley, Marina F McCowin, A Frances Armstrong, Simon E Coppard, Felipe Aguilera, Omri Bronstein, Data curation, Writing – review and editing; Andreas Kroh, Data curation, Formal analysis; Rich Mooi, Conceptualization, Data curation, Supervision, Writing – review and editing, Funding acquisition; Greg W Rouse, Conceptualization, Resources, Supervision, Writing – review and editing, Funding acquisition

## Author ORCIDs

Nicolás Mongiardino Koch (iD) http://orcid.org/0000-0001-6317-5869
Simon E Coppard (iD) http://orcid.org/0000-0002-8930-2923
Felipe Aguilera (iD) http://orcid.org/0000-0003-3235-931X
Omri Bronstein (iD) http://orcid.org/0000-0003-2620-3976
Andreas Kroh (iD) http://orcid.org/0000-0002-8566-8848

## Decision letter and Author response

Decision letter https://doi.org/10.7554/eLife.72460.sa1
Author response https://doi.org/10.7554/eLife.72460.sa2

---

## Additional files

### Supplementary files

• Transparent reporting form

### Data availability

Raw transcriptomic and genomic reads have been deposited in NCBI under Bioproject accession numbers PRJNA767441, PRJNA746411 and PRJNA746412. Assemblies, phylogenomic datasets and R code to plot chronospaces and explore the effects of methodological decisions on divergence time estimation can be found at the following Dryad Digital Repository: https://doi.org/10.5061/dryad.brv15dv9t.

The following dataset was generated:

| Author(s) | Year | Dataset title | Dataset URL | Database and Identifier |
|---|---|---|---|---|
| Mongiardino Koch N, Thompson JR, Hiley AS, McCowin MF, Armstrong AF, Coppard SE, Aguilera F, Bronstein O, Kroh A, Mooi R, Rouse GW | 2022 | Data from: Phylogenomic analyses of echinoid diversification prompt a re-evaluation of their fossil record | https://dx.doi.org/10.5061/dryad.brv15dv9t | Dryad Digital Repository, 10.5061/dryad.brv15dv9t |

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

## Appendix 1

### Sequencing and further bioinformatic details

*Apatopygus recens*, *Aspidodiadema hawaiiense*, *Fellaster zelandiae*, *Histocidaris variabilis*, *Rhyncholampas pacificus*, *Sinaechinocyamus mai*, *Stereocidaris neumayeri*, *Tromikosoma hispidum*. Tissue subsamples were finely chopped with a scalpel and preserved in RNA*later* (Ambion) buffer solution for 1 day at 4°C to allow the RNA*later* to effectively penetrate the tissues, followed by long-term storage at –80°C until RNA extraction. Total RNA was extracted from 1.5 mm Triple-Pure High Impact Zirconium beads (Benchmark Scientific) in Trizol (Ambion), using Direct-zol RNA Miniprep Kit (Zymo Research) with in-column DNase I incubation to remove genomic DNA. Prior to mRNA capture, total RNA concentration was estimated using Qubit RNA HS Assay Kit (Invitrogen; range = 8.33–96 ng/µl), and quality was assessed using either High Sensitivity RNA ScreenTape or RNA ScreenTape with an Agilent 4200 TapeStation. Mature mRNA was isolated from total RNA and libraries were prepared using KAPA mRNA HyperPrep Kit (KAPA Biosystems) following the manufacturer's instructions (including sample customization based on total RNA quantity and quality values), targeting an insert size circa 500 base pairs (bp), and using custom 10-nucleotide Illumina TruSeq style adapters (*Faircloth and Glenn, 2012*). Post-amplification, DNA concentration was estimated using Qubit dsDNA BR Assay Kit (Invitrogen; range = 6.06–13.4 ng/µl). Concentration, quality, and molecular weight distribution of libraries (range = 547–978 bp) were also assessed using Genomic DNA ScreenTape with an Agilent 4200 TapeStation. Ten libraries (including two annelids not employed here) were sequenced on one lane of a multiplexed run using NovaSeq S4 platform with 100 bp paired-end reads at the IGM Genomics Center (University of California San Diego).

The assembled transcriptomes were sanitized from cross-contaminant reads product of multiplexed sequencing using CroCo v. 1.1 (*Simion et al., 2018*). These eight datasets, as well as two annelid transcriptomes not employed in this study but sequenced in the same lane, were incorporated. Transcripts considered over- or under-expressed across samples (as defined by default parameters) were kept. This resulted in an average removal of 1.26% of assembled transcripts (range: 0.4% for *Aspidodiadema* to 3.06% for *Histocidaris*).

*Clypeaster japonicus*, *Encope emarginata*, *Leodia sexiesperforata*, *Peronella japonica*. Eggs were collected from a single female specimen of each species and immediately placed into RNA later for preservation. The samples were left in RNAlater at 4°C for at least 1 day and then transferred to a –80°C freezer until RNA extraction. RNA extraction was performed using Trizol and was then treated with Ambion's Turbo DNA-free kit. RNA was quantified using both Nanodrop and an Agilent 2100 Bioanalyzer. Only RNA samples with an RNA integrity number (RIN) of >7 were used. RNA-Seq libraries were prepared using the NEB Ultra Directional kit using the standard protocol and multiplexed with 6 bp NEB multiplex primers. Paired-end 100 libraries were generated from the resultant libraries at the UC Berkeley Genome Center using an Illumina HighSeq 2500.

*Bathysalenia phoinissa*, *Micropyga tuberculata*: The material was collected by the deep-sea cruise BIOMAGLO (*Corbari et al., 2017*) conducted in 2017 jointly by the French National Museum of Natural History (MNHN) as part of the Tropical Deep-Sea Benthos program, the French Research Institute for Exploitation of the Sea (IFREMER), the 'Terres Australes et Antarctiques Françaises' (TAAF), the Departmental Council of Mayotte and the French Development Agency (AFD), with the financial support of the European Union (Xe FED). Specimens were sorted into taxonomic bins at class level and collectively fixed in 70% ethanol at room temperature. Following taxonomic identification, tissue samples were obtained using sterilized forceps and disposable scalpel blades. Tissue subsamples were collected in high-purity 96% ethanol and stored at –40°C until DNA extraction. Total genomic DNA was extracted using the DNeasy Blood and Tissue Kit (QIAGEN) following the manufacturer's instructions, complemented by the addition of 4 µl RNAse A (QIAGEN, 100 mg/ml) for RNA digestion. Elution buffer volume was reduced to 60 µl and the elution step repeated twice per sample in order to maximize DNA concentration and yield. After collection of the first elution, spin-column membranes were rinsed again twice with fresh elution buffer (40 µl) to further increase yield. Elutions were collected separately, with the first one used for sequencing and the second for quality control. For the latter, a portion of the mitochondrial cytochrome c oxidase subunit I was amplified using PCR. Amplifications were conducted with TopTaq DNA Polymerase (QIAGEN) using 1 µl of extracted genomic DNA (approx. 10–15 ng). Details on primers and protocols can be found in *Bronstein et al., 2019*. PCR products were visualized on a 1% agarose gel, purified using ExoSAP-IT (Affymetrix), and sequenced at Microsynth GmbH (Vienna,

Austria) with the same primers. Sequences are deposited in GenBank under the accession numbers MZ568824 and MZ568825.

Prior to library preparation, total genomic DNA concentration was estimated using Qubit DNA BR Assay Kit (Invitrogen) in order to determine library preparation strategy. This step was carried out by Macrogen (Korea), using an Illumina TruSeq DNA PCR-free kit (for *Micropyga*) and an Illumina TruSeq Nano DNA kit (for *Bathysalenia*). Library preparation followed the manufacturer's instructions, targeting an insert size of 350 bp. The libraries were sequenced by Macrogen (Korea) on a shared lane of a multiplexed run using an Illumina HiSeq X instrument with 150 bp paired-end reads. Sequencing depth was 134.1 and 161.3 million reads for *Bathysalenia* and *Micropyga*, respectively. Pre-processing of Illumina shotgun data was carried out by removal of adapters using BBDuk (https://sourceforge.net/projects/bbmap/) with settings: `minlen = 50 ktrim = r k = 23 mink = 11 tpe tbo`; followed by quality trimming of reads using UrQt v. 1.0.18 (***Modolo and Lerat, 2015***), discarding regions with phred quality score <28 and sequences of size <50. Assembly of the data was carried out with MEGAHIT v. 1.1.2 (***Li et al., 2015***), in an iterative approach followed by assessment of assembly quality using BUSCO v. 3.0.2. scores (***Seppey, 2019***) using the metazoan dataset ODB v. 9 (***Kriventseva et al., 2015***). Final BUSCO scores of the draft assemblies were 28.4% (S:27.8%, D:0.6%, F:48.2%, M:23.4%, n:978) for *Bathysalenia* and 17.2% (S:16.8%, D:0.4%, F:59.0%, M:23.8%, n:978) for *Micropyga*. As recommended by ***Hoff and Stanke, 2019***, draft genomes were masked with RepeatMasker v. 4.1.0 (***Smit et al., 2015***) prior to gene prediction, using settings: `-norna -xsmall`; resulting in 3.82% and 2.9% masked bases for *Bathysalenia* and *Micropyga*, respectively. Gene prediction was carried out using AUGUSTUS v. 3.2.3 (***Stanke et al., 2006***) using settings: `--strand= both --singlestrand= false --genemodel= partial --codingseq= on --sample = 0 --alternatives-from-sampling= false --exonnames= on --softmasking= on`. This step relied on a training dataset composed of USCOs derived from the most recent *S. purpuratus* genome (Spur v. 5).

*Echinothrix calamaris*, *Tripneustes gratilla*. Tissue samples were taken from live sea urchins housed in laboratory aquaria and total RNA was immediately extracted from 150 mg of tissue using a PureLink RNA extraction kit (Invitrogen). The quality of total RNA was assessed on a BioAnalyzer 2100 (Agilent Technologies, Santa Clara, CA) to ensure a RIN > 9 for all samples. Mature mRNA was extracted from 1 μg of total RNA and cDNA libraries were constructed using a TruSeq kit (Illumina). Quality control of libraries was assessed on a BioAnalyzer and quantification measured using qPCR. *NEBNext* Multiplex adaptors were added via ligation, and the cDNA libraries were sequenced at Genome Quebec, McGill University. Three libraries were multiplex sequenced on one lane of Illumina at a concentration of 8 pM per cDNA library using HiSeq 2500 transcriptome sequencing to generate 125 bp paired-end reads. This resulted in approximately 80 million reads for each transcriptome.

*Tetrapygus niger*. Adult specimens were collected and transported alive to the University of Concepcion. Tissue samples (female gonad and tube feet) from one individual were finely chopped with a scalpel, and total RNA was extracted immediately using TriReagent (Sigma), following the manufacturer's instructions. Total RNA concentration was estimated using QuantiFluor RNA System (111.84–339.04 ng/μl) and quality was assessed using Agilent RNA 6000 Nano kit with an Agilent 2100 TapeStation (RIN: 5.4–8.4). cDNA libraries were prepared at Genoma Mayor SpA (Chile) using TruSeq Stranded mRNA Kit (Illumina) and sequenced on a multiplexed run using Illumina HiSeq 4000 platform with 150 paired-end reads, resulting in 54.1 M reads for gonad female and 52.7 M reads for tube feet, respectively. These two transcriptomic datasets were combined before performing all steps of bioinformatic processing.

*Eucidaris metularia*. A single specimen was preserved in RNA*later* after being starved overnight and kept for 1 day at 4°C before long-term storage at –80°C. Total RNA was extracted using Direct-zol RNA Miniprep Kit (with in-column DNase treatment; Zymo Research) from Trizol. mRNA was isolated with Dynabeads mRNA Direct Micro Kit (Invitrogen). RNA concentration was estimated using Qubit RNA broad range assay kit, and quality was assessed using RNA ScreenTape with an Agilent 4200 TapeStation. Values were used to customize downstream protocols following manufacturers' instructions. Library preparation was performed with KAPA-Stranded RNA-Seq kits, targeting an insert size in the range of 200–300 bp. The quality and concentration of libraries were assessed using DNA ScreenTape. The library was sequenced in a multiplexed pair-end runs using Illumina HiSeq

4000 with seven other libraries in the same lane (all previously published in *Mongiardino Koch et al., 2018*). In order to minimize read crossover, 10 bp sequence tags designed to be robust to indel and substitution errors were employed (*Faircloth and Glenn, 2012*).

The assembled transcriptome was filtered from cross-contaminant reads product of multiplexed sequencing using CroCo v. 1.1 (*Simion et al., 2018*). Seven other transcriptomes sequenced together were also employed, and results of this step are reported in Fig. S1 of *Mongiardino Koch and Thompson, 2021d*.

**Appendix 1—table 1.** Transcriptomic/genomic datasets added in this study relative to the taxon sampling of *Mongiardino Koch et al., 2018* and *Mongiardino Koch and Thompson, 2021d*. Taxa with citations were taken from the literature, and details can be found in the corresponding studies and associated NCBI BioProjects. Taxa are shown in alphabetical order; those identified with 'OG' are outgroup taxa (i.e., non-echinoids). Voucher specimens are deposited at the Benthic Invertebrate Collection, Scripps Institution of Oceanography (SIO-BIC), and the Echinoderm Collection, Muséum National d'Histoire Naturelle (MNHN-IE). If multiple accession numbers are shown for a given taxon, these datasets were merged before assembly. Similar details for all other specimens can be found in *Mongiardino Koch et al., 2018* and *Mongiardino Koch and Thompson, 2021d*.

| Taxon | Citation | Tissue type | Collector | Locality (depth) | Voucher | GenBank accession number |
|---|---|---|---|---|---|---|
| *Amphiura filiformis* (OG) | *Dylus et al., 2016* | – | – | – | – | SRX2255774 |
| *Apatopygus recens* | This study | Mixed | Owen Anderson | Foveaux Strait, South Island, New Zealand | SIO-BIC E7142 | SRR16134561 |
| *Aspidodiadema hawaiiense* | This study | Mixed | Greg Rouse, Avery Hiley | Seamount 4, Costa Rica, Pacific Ocean (1003 m) | SIO-BIC E7363 | SRR16134560 |
| *Asterias rubens* (OG) | *Reich et al., 2015* | – | – | – | – | SRX445860 |
| *Astrophyton muricatum* (OG) | *Janies et al., 2016*; *Linchangco et al., 2017* | – | – | – | – | SRX1391908 |
| *Bathysalenia phoinissa* | This study | Tube feet, spine muscles | BIOMAGLO Cruise Team | Mozambique Channel, Mayotte (295–336 m) | MNHN-IE-2016–23 | SRR15130003 |
| *Clypeaster japonicus* | This study | Eggs | Frances Armstrong | Misaki Marine Biological Station, Kanagawa, Japan | – | SRR16134552 |
| *Echinothrix calamaris* | This study | Male gonad | – | Philippines | – | SRR16134551 |
| *Encope emarginata* | This study | Eggs | Gulf Specimen Marine Lab | Apalachee Bay, FL, USA | – | SRR16134550 |
| *Eucidaris metularia* | This study | Mixed | Greg Rouse | Al-Fahal Reef, Red Sea, Makkah, Saudi Arabia | SIO-BIC 2017–008 | SRR16134549 |
| *Fellaster zelandiae* | This study | Spines, tube feet, gut, joining tissue of Aristotle's lantern | Wilma Blom | Western end of Cornwallis Beach, Auckland, New Zealand | SIO-BIC E7920 | SRR16134548 |
| *Histocidaris variabilis* | This study | Gonad | Greg Rouse, Avery Hiley | Las Gemelas Seamount, near Isla del Coco, Costa Rica, Pacific Ocean (571 m) | SIO-BIC E7350 | SRR16134547 |

*Appendix 1—table 1 Continued on next page*

*Appendix 1—table 1 Continued*

| Taxon | Citation | Tissue type | Collector | Locality (depth) | Voucher | GenBank accession number |
|---|---|---|---|---|---|---|
| *Lamberticrinus messingi* (OG) | *Clouse et al., 2015*; *Janies et al., 2016* | – | – | – | – | SRX1397823 |
| *Leodia sexiesperforata* | This study | Eggs | Frances Armstrong | Bocas del Toro, Panama | – | SRR16134546 |
| *Metacrinus rotundus* (OG) | *Koga et al., 2016* | – | – | – | – | DRX021520 |
| *Micropyga tuberculata* | This study | Tube feet, spine muscles | BIOMAGLO Cruise Team | Mozambique Channel, Îles Glorieuses (385–410 m) | MNHN-IE-2016–39 | SRR15130004 |
| *Ophiocoma echinata* (OG) | *Reich et al., 2015* | – | – | – | – | SRX445856 |
| *Peronella japonica* | This study | Eggs | Frances Armstrong | Kanazawa, Japan | – | SRR16134545 |
| *Rhyncholampas pacificus* | This study | Mixed | Carlos A. Conejeros-Vargas | Panteón Beach, Puerto Ángel Bay, Oaxaca, Mexico | SIO-BIC E7918 | SRR16134558 |
| *Scaphechinus mirabilis* | *Simakov et al., 2015* | – | – | – | – | DRX187887 DRX187888 |
| *Sinaechinocyamus mai* | This study | Gregory's diverticulum | Jih-Pai Lin | Miaoli County, Taiwan | SIO-BIC E7919 | SRR16134557 |
| *Sterechinus neumayeri* | *Collins et al., 2021* | – | – | – | – | ERX3498697 ERX3498698 ERX3498699 ERX3498700 ERX3498701 |
| *Stereocidaris nascaensis* | This study | Muscle surrounding Aristotle's lantern | Charlotte Seid | Off San Ambrosio, Desventuradas Islands, Chile (215 m) | SIO-BIC E7154 | SRR16134559 |
| *Tetrapygus niger* | This study | Female gonad, tube feet | Felipe Aguilera | Dichato Bay, Chile | – | SRR16134553 SRR16134554 |
| *Tripneustes gratilla* | This study | Pedicellariae | – | Philippines | – | SRR16134556 |
| *Tromikosoma hispidum* | This study | Tube feet | Lisa Levin, Todd Litke | Quepos Plateau, Costa Rica, Pacific Ocean (2067 m) | SIO-BIC E7251 | SRR16134555 |

**Appendix 1—table 2.** Details of molecular datasets and supermatrix.

Statistics for raw reads and assemblies are shown for datasets incorporated in this study relative to the sampling of *Mongiardino Koch et al., 2018* and *Mongiardino Koch and Thompson, 2021d*, where similar statistics can be found for the other datasets. Taxa are shown in alphabetical order; those identified with 'OG' are outgroup taxa (i.e., non-echinoids). Novel datasets correspond to Illumina pair-end transcriptomes, except for two draft genomes (*Bathysalenia phoinissa* and *Micropyga tuberculata*). Mean insert size is expressed in number of base pairs. For transcriptomes, read pairs (initial) shows numbers input into Agalma (*Dunn et al., 2013*), that is, after processing with Trimmomatic (*Bolger et al., 2014*) or UrQt (*Modolo and Lerat, 2015*), while read pairs (retained) show those that passed the Agalma curation checks (including ribosomal, adapter, quality, and compositional filters), and represent the final number of read pairs used for assembly. For genomes, see information in the bioinformatic details above. Assemblies were further sanitized with either alien_index (*Ryan, 2014*) alone or in combination with CroCo (*Simion et al., 2018*), and the

number of assembled transcripts/contigs shows the size of datasets after these curation steps. The number of loci shows the occupancy of terminals in the supermatrix output by Agalma (1346 loci at 70% occupancy), after which loci were further removed by TreeShrink (*Mai and Mirarab, 2018*), resulting in the final occupancy scores.

| Species | Mean insert size | Read pairs (initial) | Read pairs (retained) | Assembled transcripts/ contigs | Number of loci | Removed by TreeShrink | Final occupancy |
|---|---|---|---|---|---|---|---|
| *Abatus agassizii* | – | – | – | – | 522 | 5 | 38.4 |
| *Abatus cordatus* | – | – | – | – | 497 | 2 | 36.8 |
| *Acanthaster planci* (OG) | – | – | – | – | 864 | 8 | 63.6 |
| *Amphiura filiformis* (OG) | 180.1 | 61,558,173 | 52,206,727 | 416,946 | 761 | 8 | 55.9 |
| *Apatopygus recens* | 415.3 | 74,315,687 | 56,898,850 | 274,380 | 1152 | 5 | 85.2 |
| *Apostichopus japonicus* (OG) | – | – | – | – | 849 | 9 | 62.4 |
| *Araeosoma leptaleum* | – | – | – | – | 1184 | 2 | 87.8 |
| *Arbacia lixula* | – | – | – | – | 1234 | 1 | 91.6 |
| *Aspidodiadema hawaiiense* | 228.7 | 109,716,219 | 104,518,714 | 311,032 | 1060 | 11 | 77.9 |
| *Asterias rubens* (OG) | 230.4 | 31,890,613 | 25,495,009 | 103,090 | 805 | 9 | 59.1 |
| *Asthenosoma varium* | – | – | – | – | 1254 | 7 | 92.6 |
| *Astrophyton muricatum* (OG) | 195.9 | 25,191,954 | 22,281,829 | 149,146 | 478 | 5 | 35.1 |
| *Bathysalenia phoinissa* | – | – | – | 154,120 | 605 | 6 | 44.5 |
| *Brissus obesus* | – | – | – | – | 773 | 0 | 57.4 |
| *Caenopedina hawaiiensis* | – | – | – | – | 1094 | 2 | 81.1 |
| *Clypeaster japonicus* | 198.6 | 10,505,520 | 9,298,316 | 74,743 | 829 | 1 | 61.5 |
| *Clypeaster rosaceus* | – | – | – | – | 1242 | 2 | 92.1 |
| *Clypeaster subdepressus* | – | – | – | – | 1233 | 3 | 91.4 |
| *Colobocentrotus atratus* | – | – | – | – | 1158 | 0 | 86.0 |
| *Conolampas sigsbei* | – | – | – | – | 1001 | 5 | 74.0 |
| *Cystechinus* c.f. *giganteus* | – | – | – | – | 661 | 0 | 49.1 |
| *Dendraster excentricus* | – | – | – | – | 337 | 1 | 25.0 |
| *Diadema setosum* | – | – | – | – | 305 | 1 | 22.6 |
| *Echinarachnius parma* | – | – | – | – | 1187 | 3 | 88.0 |
| *Echinocardium cordatum* | – | – | – | – | 1012 | 2 | 75.0 |
| *Echinocardium mediterraneum* | – | – | – | – | 1185 | 1 | 88.0 |
| *Echinocyamus crispus* | – | – | – | – | 709 | 6 | 52.2 |
| *Echinometra mathaei* | – | – | – | – | 1055 | 0 | 78.4 |
| *Echinothrix calamaris* | 203.9 | 68,871,087 | 60,443,904 | 251,788 | 1252 | 4 | 92.7 |
| *Encope emarginata* | 206.2 | 12,015,888 | 10,461,870 | 66,241 | 1076 | 1 | 79.9 |
| *Eucidaris metularia* | 321.6 | 38,802,159 | 29,047,401 | 83,318 | 412 | 2 | 30.5 |
| *Eucidaris tribuloides* | – | – | – | – | 414 | 0 | 30.8 |
| *Evechinus chloroticus* | – | – | – | – | 1196 | 1 | 88.8 |

*Continued on next page*

*Continued*

| Species | Mean insert size | Read pairs (initial) | Read pairs (retained) | Assembled transcripts/ contigs | Number of loci | Removed by TreeShrink | Final occupancy |
|---|---|---|---|---|---|---|---|
| *Fellaster zelandiae* | 313.9 | 62,619,791 | 51,742,748 | 168,598 | 1269 | 1 | 94.2 |
| *Heliocidaris erythrogramma* | – | – | – | – | 931 | 0 | 69.2 |
| *Histocidaris variabilis* | 329.2 | 84,103,672 | 73,884,180 | 144,044 | 1189 | 5 | 88.0 |
| *Holothuria forskali* (OG) | – | – | – | – | 743 | 8 | 54.6 |
| *Lamberticrinus messingi* (OG) | 195.0 | 22,989,820 | 20,234,597 | 59,049 | 351 | 3 | 25.9 |
| *Leodia sexiesperforata* | 203.8 | 16,211,182 | 14,174,475 | 68,964 | 1064 | 1 | 79.0 |
| *Leptosynapta clarki* (OG) | – | – | – | – | 403 | 4 | 29.6 |
| *Lissodiadema lorioli* | – | – | – | – | 770 | 0 | 57.2 |
| *Loxechinus albus* | – | – | – | – | 1265 | 1 | 93.9 |
| *Lytechinus variegatus* | – | – | – | – | 1257 | 2 | 93.2 |
| *Mellita tenuis* | – | – | – | – | 1002 | 1 | 74.4 |
| *Meoma ventricosa* | – | – | – | – | 1053 | 0 | 78.2 |
| *Mesocentrotus nudus* | – | – | – | – | 1247 | 0 | 92.6 |
| *Metacrinus rotundus* (OG) | 198.5 | 23,791,832 | 20,900,345 | 83,718 | 642 | 7 | 47.2 |
| *Micropyga tuberculata* | – | – | – | 170,514 | 415 | 4 | 30.5 |
| *Ophiocoma echinata* (OG) | 278.5 | 28,427,026 | 24,836,025 | 130,153 | 712 | 7 | 52.4 |
| *Paracentrotus lividus* | – | – | – | – | 1266 | 0 | 94.1 |
| *Patiria miniata* (OG) | – | – | – | – | 740 | 8 | 54.4 |
| *Peronella japonica* | 208.3 | 17,696,287 | 15,707,931 | 106,110 | 1043 | 6 | 77.0 |
| *Prionocidaris baculosa* | – | – | – | – | 794 | 3 | 58.8 |
| *Psammechinus miliaris* | – | – | – | – | 1173 | 1 | 87.1 |
| *Rhyncholampas pacificus* | 346.8 | 63,116,413 | 52,160,741 | 234,910 | 1070 | 2 | 79.3 |
| *Saccoglossus kowalevskii* (OG) | – | – | – | – | 668 | 6 | 49.2 |
| *Scaphechinus mirabilis* | 401.7 | 10,169,195 | 8,796,067 | 127,664 | 974 | 4 | 72.1 |
| *Sinaechinocyamus mai* | 297.2 | 60,208,172 | 52,265,201 | 164,646 | 1233 | 1 | 91.5 |
| *Sphaerechinus granularis* | – | – | – | – | 1214 | 1 | 90.1 |
| *Sterechinus neumayeri* | 207.3 | 26,376,279 | 21,308,840 | 122,001 | 1097 | 1 | 81.4 |
| *Stereocidaris nascaensis* | 327.0 | 69,962,495 | 60,316,050 | 121,508 | 1200 | 6 | 88.7 |
| *Stomopneustes variolaris* | – | – | – | – | 908 | 0 | 67.5 |
| *Strongylocentrotus purpuratus* | – | – | – | – | 1266 | 5 | 93.7 |
| *Tetrapygus niger* | 202.5 | 63,040,541 | 52,761,671 | 163,084 | 1287 | 2 | 95.5 |
| *Tripneustes gratilla* | 154.6 | 62,962,239 | 57,015,692 | 130,376 | 1309 | 1 | 97.2 |
| *Tromikosoma hispidum* | 300.4 | 57,208,357 | 47,909,038 | 234,502 | 1238 | 7 | 91.5 |

# Appendix 2

## Fossil constraints

All clade names used below refer to the corresponding crown groups.

Ambulacraria (Echinodermata-Hemichordata divergence) – **Younger bound**: 518 Ma, Middle Atdabanian (age 3, Series 3 of Cambrian). **Older bound**: 636.1 Ma, Lantian Biota, Ediacaran. **Reference:** *Benton et al., 2015*. **Notes:** The root of our tree is constrained with a younger bound concurrent with the earliest occurrence of stereom in the fossil record (see below). The older bound is based on the maximum age of the Lantian Biota, a Lagerstätte lacking any trace of eumetazoan fossils.

Echinodermata (Pelmatozoan-Eleutherozoan divergence) – **Younger bound**: Unconstrained. **Older bound:** 515.5 Ma, Middle Atdabanian (age 3, Series 3 of Cambrian). **Reference**: *Benton et al., 2015*. **Notes:** This divergence (which represents the divergence of crown group echinoderms) has an older bound based upon the earliest occurrence of stereom in the fossil record. Stereom constitutes an echinoderm synapomorphy readily recognizable in the fossil record due to its unique mesh-like structure (*Bottjer et al., 2006*). The first records of stereom point to a sudden and global appearance within the middle Atdabanian (*Zamora et al., 2013*). Note that this is also used as younger bound for the divergence between echinoderms and hemichordates.

Echinozoa (Echinoidea-Holothuroidea divergence) – **Younger bound:** 469.96 Ma, Top of *Pseudoclamacograptus decorates* graptolite zone. **Older bound:** 461.95 Ma, Top Floian. **Reference:** *Cooper and Sadler, 2012*; *Erkenbrack, 2019*. **Notes:** The oldest crown eleutherozoan fossils are disarticulated holothurian calcareous ring elements which are known from the Red *Orthoceras* limestone of Sweden. This falls within the *P. decorates* graptolite zone.

Asterozoa (Ophiuroidea-Asteroidea divergence) – **Younger bound**: 521 Ma, Top of *Psigraptus jacksoni* Zone. **Older bound**: 480.55 Ma, Base of Series 2 Cambrian Period. **Reference**: *Cooper and Sadler, 2012*; *Erkenbrack, 2019*. **Taxon**: *Maydenia roadsidensis*. **Notes**: The divergence of asterozoan classes is calibrated based upon the stratigraphic occurrence of *Maydenia roadsidensis*, the oldest asterozoan, which is a stem group ophiuroid (*Hunter and Ortega-Hernández, 2021*) and occurs in the *Psigraptus jacksoni* Zone of the Floian.

Asteroidea – **Younger bound:** 239.10 Ma, Top Fassanian, Ladinian, Mo3, *nodosus* biozone, Triassic. **Older bound**: 252.16 Ma, P-T Boundary. **Reference**: *Villier et al., 2017*. **Taxon**: *Trichasteropsis weissmanni*. **Notes**: Fossil evidence suggests the divergence of the asterozoan crown group took place sometime in the Early or Middle Triassic (*Villier et al., 2017*). Based upon its phylogenetic position and stratigraphic occurrence as the oldest crown group asteroid, we use the forcipulatacean *T. weissmanni* from the Middle Triassic Muschelkalk as the soft bound on this divergence.

Crinoidea – **Younger bound**: 247.06 Ma, Top Spathian (Paris Biota, Lower Shale unit, Thaynes Group, Spathian, Triassic). **Older bound**: Unconstrained. **Reference**: *Saucède et al., 2019*. **Taxon**: *Holocrinus*. **Notes**: Crown group Crinoidea is difficult to define, though a rapid post-Paleozoic morphological diversification is supported by fossil evidence. We use *Holocrinus* from the Early Triassic of the Thaynes group to calibrate the younger bound on the divergence of the crown group (i.e., the split between Isocrinida and all other extant crinoids).

Ophiuroidea (Euryophiurida-Ophintegrida divergence) – **Younger bound**: 247.06 Ma, Top Spathian (Paris Biota, Lower Shale unit, Thaynes Group, Spathian, Triassic). **Older bound**: Unconstrained. **Reference**: *Thuy and Escarguel, 2019*. **Taxon**: *Shoshonura brayardi*. **Notes**: *S. brayardi* from the Spathian Thaynes Group of the Western USA is a member of the crown group ophiuroid clade Ophiodermatina. It is thus the currently oldest described member of the ophiuroid crown group, setting its minimum age of divergence.

Pneumonophora (Holothuriida-Neoholothuriida divergence) – **Younger bound**: 259.8 Ma, S*pinosus* zone of Early Ladinian (used base Ladinian). **Older bound**: 241.5 Ma, Base Wuchiapingian. **Reference**: *Miller et al., 2017*; *Erkenbrack, 2019*. **Notes**: The oldest (undescribed) holothuriid calcareous ring ossicles are from the *Spinosus* zone of the Ladinian of Germany and calibrate its divergence from neoholothuriids.

Echinoidea (Cidaroidea-Euechinoidea divergence) – **Younger bound**: 298.9 Ma, Base Carnian. **Older bound**: 237 Ma, Base Permian. **Reference**: *Kroh, 2011*. **Taxon**: *Triassicidaris? ampezzana*. **Notes**: The recent phylogenetic analysis of *Mongiardino Koch and Thompson, 2021d* found

many traditional early crown group echinoids as members of the stem group. Given this result, *Triassicidaris? ampezzana*, a cidaroid which is known from the St Cassian beds of Italy, becomes the current oldest crown group echinoid.

Pedinoida-Aspidodiadematoida – **Younger bound**: 209.46 Ma, Top of Norian. **Older bound**: 252.16 Ma, P-T boundary. **Reference**: See discussion in supplement of *Thompson et al., 2017a*. **Taxon**: *Diademopsis* ex. gr. *heberti*. **Notes**: The oldest euechinoid, *Diademopsis* ex. gr. *heberti*, which is a pedinoid, is known from the Norian of Peru. This fossil calibrates the younger bound on the pedinoid-aspidodiadematoid divergence.

Carinacea (Echinacea-Irregularia divergence) – **Younger bound**: 234.5 Ma, Top Sinemurian. **Older bound**: 190.8 Ma, Top Julian one ammonoid zone of Norian. **Reference**: *Li et al., 2020*. **Taxon**: *Jesionekechinus*. **Notes**: The divergence of Carinacea is calibrated based upon the oldest irregular echinoid, *Jesionekechinus*, which occurs above the Sinemurian of New York Canyon, Nevada. For a more detailed discussion of this divergence, see the supplement of *Thompson et al., 2017a*.

Salenioida-(Camarodonta + Stomopneustoida) divergence – **Younger bound**: 228.35 Ma, Base Hettangian. **Older bound**: 201.3 Ma, Top Carnian. **Reference**: *Smith et al., 2006*. **Taxon**: *Acrosalenia chartroni*. **Notes**: The stem group salenioid *Acrosalenia chartroni* from the Hettangian of France is the earliest representative of Echinacea. Given the topology recovered by our analyses, this fossil was used to calibrate the divergence between salenioids and the clade composed of stomopneustoids and camarodonts.

Stomopneustoida-Camarodonta divergence – **Younger bound**: 201.3 Ma, Top Pliensbachian. **Older bound**: 182.7 Ma, Base Jurassic. **Reference**: See discussion in supplement of *Thompson et al., 2017a*. **Taxon**: Stomechinids from Morocco like *Diplechinus hebbriensis* (*Smith, 2010*). **Notes**: The oldest stomechinids, which are stem group stomopneustoids, are from the Early Jurassic of Morocco.

Neognathostomata-Atelostomata divergence – **Younger bound**: 234.5 Ma, Base of Toarcian. **Older bound**: 182.7 Ma, St Cassian Beds, bottom of Julian two ammonoid Zone. **Reference**: *Smith et al., 2006*; *Li et al., 2020*. **Taxon**: Younger bound is set by *Galeropygus sublaevis* (older bound is relatively uninformative). **Notes**: The base of the Toarcian (which contains the oldest neognathostomate *G. sublaevis*) is used as the lower bound on the divergence between Neognathostomata and all other crown irregular echinoids. Note that depending on the position of the unsampled echinoneoids, this split might also correspond to the origin of crown group Irregularia; see recent topologies recovered by *Lin et al., 2020* and *Mongiardino Koch and Thompson, 2021d*.

Atelostomata (Holasteroida-Spatangoida divergence) – **Younger bound**: 137.68 Ma, Bottom of *Campylotoxus* zone in Valanginian. **Older bound**: 201.3 Ma, Base of Jurassic. **Reference**: *François et al., 2003*. **Taxon**: Younger bound is based on *Toxaster* (older bound is loosely uninformative). **Notes**: The toxasterids are stem group spatangoids that calibrate the divergence between spatangoids and holasteroids. The oldest *Toxaster* are from the *Campylotoxus* zone in the Valanginian, and set the younger bound on the divergence.

Echinolampadacea (Cassiduloida-Scutelloida divergence) – **Younger bound**: 113 Ma, Base Albian. **Older bound**: 145 Ma, Base Cretaceous. **Reference**: *Cooke, 1955*. **Taxon**: Younger bound is based on *Eurypetalum rancheriana* (older bound is loosely uninformative). **Notes**: The oldest member of Echinolampadacea is the faujasiid *Eurypetalum rancheriana* – originally placed in the genus *Faujasia* and later transferred (*Kier, 1962*; *Souto et al., 2019*) – from the Late Albian of Colombia.

Scutelloida (Scutelliformes-Laganiformes divergence) – **Younger bound**: 56.0 Ma, Bottom of Eocene. **Older bound**: Unconstrained. **Reference**: *Roman, 1989*; *Smith et al., 2006*. **Taxon**: Younger bound is based on *Eoscutum doncieuxi*. **Notes**: We use the base of the Eocene as the younger bound on the divergence between scutelliforms and laganiforms. This is based upon the occurrence of *E. doncieuxi*, the oldest scutelliform, which is known from the Lower Eocene. The older bound is left unconstrained in order to account for the current uncertainty in the origin of the clade (i.e., allow for Mesozoic ages).

Leodia-(Encope + Mellita) divergence – **Younger bound**: 23.03 Ma, Base Miocene. **Older bound**: 33.9 Ma, Base of Oligocene. **Reference**: *Durham, 1994*; *Coppard and Lessios, 2017*. **Taxon**: Younger bound is based on *Encope michoacanensis* (older bound is loosely uninformative). **Notes**: This node is constrained using the oldest representative of these three genera of mellitids.

Clypeasteroida – **Younger bound**: 47.8 Ma, Base Lutetian. **Older bound**: Unconstrained. **Reference**: *Via and Padreny, 1970*; *Ali, 1983*. **Taxon**: Younger bound is based on *Clypeaster calzadai* and *Clypeaster moianensis*. **Notes**: The oldest known fossil species of Clypeasteroida (the clypeasterids *C. calzadai* and *C. moianensis*) are from the middle Eocene, upper Lutetian (Biarritzian) of Cataluña, Spain. The older bound is left unconstrained in order to account for the current uncertainty in the origin of the clade (i.e., allow for Mesozoic ages).

Strongylocentrotidae-Toxopneustidae divergence – **Younger bound**: 44.4 Ma, Eocene Planktonic Zones E9-E12. **Older bound**: 56.0 Ma, Base Eocene. **Reference**: *Wade et al., 2011*; *Gold et al., 2018*. **Taxon**: Younger bound is based on *Lytechinus axiologus* (older bound is loosely uninformative). **Notes**: The oldest member of the clade comprising toxopneustids and strongylocentrotids is the toxopneustid *Lytechinus axiologus*. This taxon is known from the Eocene Yellow Limestone Group of Jamaica. The Yellow Limestone group represents the Eocene Planktonic Zones E9-E12.

# Appendix 3

## Supplementary figures

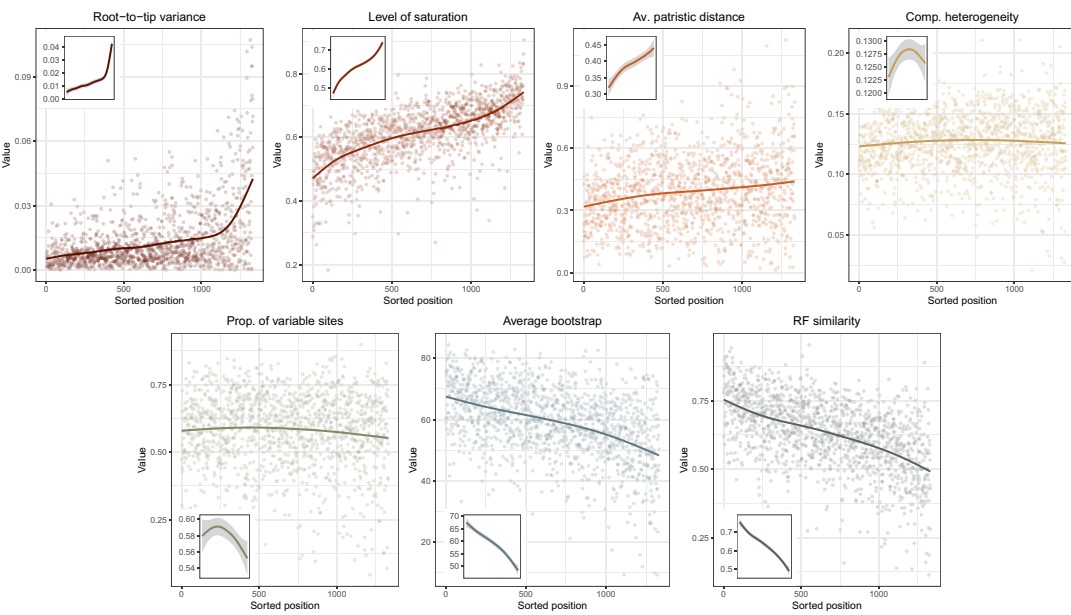

**Appendix 3—figure 1.** Ordering of loci enforced using *genesortR* (***Mongiardino Koch, 2021b***) and its relationship to the seven gene properties employed. High ranking loci (i.e., the most phylogenetically useful) show low root-to-tip variances (or high clock-likeness), low saturation, and low compositional heterogeneity, as well as high average bootstrap and Robinson-Foulds similarity to a target topology (in this case, with the contentious relationship among major lineages of Echinacea collapsed).

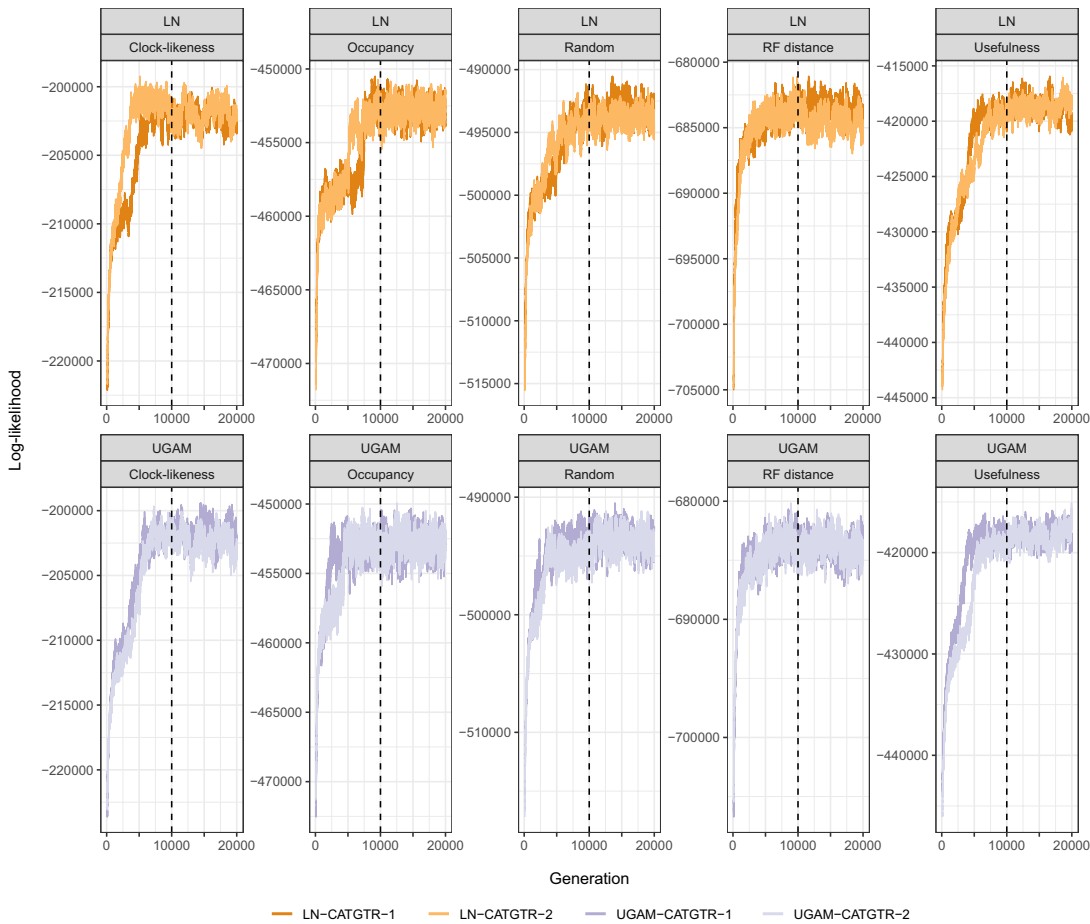

**Appendix 3—figure 2.** Trace plots of the log-likelihood of different time calibration runs. All runs show evidence of reaching convergence and stationarity before our imposed burn-in fraction of 10,000 generations (dashed lines). For simplicity, only runs under the CAT + GTR + G model and Cauchy priors are plotted. Those run under uniform node age priors behaved identically, while those run under GTR + G converged much faster.

