## [Editor Report]

The study by Mongiardino Koch et al., presents new phylogenomic and molecular clock analyses of echinoids. The study uses state of the art phylogenetic approaches and includes 18 newly sequenced genomes and transcriptomes, which are used to estimate the tree topology and divergence times of major groups of echinoids. The molecular clock-estimated times of origin of particular echinoid lineages predate the lineages' appearance on the fossil record by tens of millions of years, prompting re-evaluation of the early evolution of echinoid diversity.

---

## [Decision Letter]

**Decision letter after peer review:**

Thank you for submitting your article "Phylogenomic analyses of echinoid diversification prompt a re-evaluation of their fossil record" for consideration by *eLife*. Your article has been reviewed by 2 peer reviewers, and the evaluation has been overseen by a Reviewing Editor and George Perry as the Senior Editor. The following individual involved in review of your submission has agreed to reveal their identity: Iker Irisarri (Reviewer #1).

Essential revisions:

1) Explore the effect of using different prior distributions for calibrations, as suggested by Reviewer #1.

2) Tone down the importance of your chronospace approach – as is, it is highly overstated.

*Reviewer #1 (Recommendations for the authors):*

I really enjoyed reading this manuscript; it is very well written and provides a set of interesting results after a thorough exploration of the data. Congratulations! Below are some comments that I think would further strengthen the value of this nice manuscript.

I missed a short description of the new phylogenomic datasets (number of taxa, loci, total and informative positions, etc.) as well as the inference methods (ASTRAL, ML, Bayesian) at the beginning of the Results section. This would help the reader to better grasp the value and strength of the phylogenomics results directly here.

Fossil calibrations. It is a pity that the effect of using different prior distributions for calibrations was not explored. For example, PhyloBayes implements cauchy distributions that can be used to produce more informative priors than uniform distribution. I think this is one of the key parameters in molecular clock analyses and assessing its effect would deepen the understanding of the presented sensitivity analyses and provide a more comprehensive baseline for the chronospace analysis.

I found it very interesting that uncorrelated vs. autocorrelated clock models had the largest effect on inferred ages, even though we often have little (if any) evidence to select one or the other. Interestingly, similar results were obtained in a recent study of eukaryotes (Strassert et al., 2021; https://www.nature.com/articles/s41467-021-22044-z). The authors cite the tool CorrTest (ref 73) as a possibility of choosing between clocks models but do not test it on their data. I think testing branch length correlation with CorrTest could provide interesting results and maybe favor a set of younger or older ages (especially for the deep nodes) that could agree better with the fossil record.

Line 343-ff. This is more of a curiosity, but I wondered if there is evo-devo data on Aristotle’s lantern that could explain the mentioned dynamic evolutionary history in terms of different convergent origins, neoteny etc.

Methods. Results on the convergence of ExaBayes runs are not provided.

Supplement. Are the new specimens employed in transcriptome and genome sequencing associated with museum vouchers?

Figure 1. Small specimens are hardly visible even after zooming (please check resolution).

Figure 2. (A) says “Favored topology” – could you please state to which analysis the phylogram corresponds to? In line 154: please, avoid the word “significant” unless referring to statistical tests. (D) why does the legend have gradients of green, blue and red? The meaning is not clear to me.

*Reviewer #2 (Recommendations for the authors):*

I found the text long in places and thus tedious to read. Particularly the introduction and the discussion. The intro could have a tighter narrative more focused on the discrepancies of the fossil record and divergence times, and on discrepancies in topology without the needed to review so much echinoid biology. The discussion appears too long. Too much effort is made on justifying the chronospace approach. This perhaps does not need to be justified at all beyond a sentence or two.

Because the clock model has such an impact, you should explore this further. PhyloBayes provides the ability to test for the various rate models using Bayes factors and you should try this. Because this analyses are computationally expensive, you can do them on a reduced amount of data. MCMCtree also allows you to test for various clock models with Bayes factors.

Additional points:

The y-axis (Δ likelihood) of panel C in Figure 2 needs to be explained in the legend. What is the benchmark likelihood?

In the figure legends, the program/models used to infer the tree and times should be indicated. From the methods it’s clear that many many different methods were tested, but from the main text and the figure legends is not clear what is being summarised in the figures.

Please remove the p-value for the multivariate analysis in line 198. This is not a replicated stochastic experiment. You are simply changing priors and substitution models and hence the posterior changes. This is a deterministic mapping between data/prior and posterior.

Panel C of Figure 4: please mention in the legend the scale of the x-axis tickmarks (10 My?).

Lines 359 to 367 have little substance. There are no figures in this paper showing the correlation structure among branch lengths. For example, figure 4 shows stacked posterior distributions, like those in previous works, so isn’t this work a victim of its own criticism? There is indeed a correlation structure among branches and times, which is not shown, and which is discarded in the plots shown here. To emphasise this point, here is paper’s text edited: “The sensitivity of inferred ages is commonly explored by running analyses under different settings and summarizing the results in tables or by stacking chronograms in order to visualize the relative position of nodes (see for example Figure 4C here and the supplementary material figures).”

---

## [Author Response]

Essential revisions:1) Explore the effect of using different prior distributions for calibrations, as suggested by Reviewer #1.

Divergence time inference was rerun under an alternative prior distribution for the calibrated nodes, as suggested by Reviewer #1, and the results have been incorporated into the manuscript. Note however that previous results had already been performed under the prior distribution Reviewer #1 considered more appropriate.

2) Tone down the importance of your chronospace approach – as is, it is highly overstated.

A stronger emphasis on chronospaces was suggested by the editors before the manuscript was sent out to reviewers. We agree that the present version contains overstatements and have reverted parts of the manuscript back to its original version.

Reviewer #1 (Recommendations for the authors):I really enjoyed reading this manuscript; it is very well written and provides a set of interesting results after a thorough exploration of the data. Congratulations! Below are some comments that I think would further strengthen the value of this nice manuscript.I missed a short description of the new phylogenomic datasets (number of taxa, loci, total and informative positions, etc.) as well as the inference methods (ASTRAL, ML, Bayesian) at the beginning of the Results section. This would help the reader to better grasp the value and strength of the phylogenomics results directly here.

A short description of the size and composition of the matrix, as well as the diversity of inference methods implemented, is now presented at the beginning of the Results section.

Fossil calibrations. It is a pity that the effect of using different prior distributions for calibrations was not explored. For example, PhyloBayes implements auchy distributions that can be used to produce more informative priors than uniform distribution. I think this is one of the key parameters in molecular clock analyses and assessing its effect would deepen the understanding of the presented sensitivity analyses and provide a more comprehensive baseline for the chronospace analysis.

PhyloBayes uses by default Cauchy priors for calibrated nodes, and so this was already the type of prior distribution under which the previous analyses were performed. After consultation with Nicolas Lartillot, the main developer of PhyloBayes, all analyses were rerun under the uniform prior as well, which requires the use of the flag -ilb (standing for “improper lower bound”). As a result of this conversation, the user manual for PhyloBayes has been updated to make this explicit. Time calibrated runs have been doubled by the use of this new prior distribution, and include now 40 settings. Results are updated to reflect the evaluation of this fourth variable. Surprisingly, the effect of the prior distribution is minimal. This result is very surprising given previous findings, and might be a result specific to the way these priors are employed by PhyloBayes. We present the new results, while acknowledging that they should probably not be generalized to other software.

I found it very interesting that uncorrelated vs. autocorrelated clock models had the largest effect on inferred ages, even though we often have little (if any) evidence to select one or the other. Interestingly, similar results were obtained in a recent study of eukaryotes (Strassert et al., 2021; https://www.nature.com/articles/s41467-021-22044-z). The authors cite the tool CorrTest (ref 73) as a possibility of choosing between clocks models but do not test it on their data. I think testing branch length correlation with CorrTest could provide interesting results and maybe favor a set of younger or older ages (especially for the deep nodes) that could agree better with the fossil record.

We had in fact tested CorrTest in our dataset, but found autocorrelation to be either supported or rejected depending on which species tree was used (i.e., which topology was employed from among the set of species trees obtained using different models and software). In our opinion, this uncertainty reduces the usefulness of CorrTest and precludes it from serving as a basis to favor a given type of clock, and is a reason why we did not focus on the results obtained under just one clock (as suggested by Reviewer #2, see below). We had not included these results in the previous version of this study, but have now added it as both reviewers have mentioned this.

Line 343-ff. This is more of a curiosity, but I wondered if there is evo-devo data on Aristotle's lantern that could explain the mentioned dynamic evolutionary history in terms of different convergent origins, neoteny etc.

We are not aware of any evo-devo information so far gathered to shed light on the interesting evolutionary history of the jaw apparatus among sand dollars and close relatives.

Methods. Results on the convergence of ExaBayes runs are not provided.

Incorporated

Supplement. Are the new specimens employed in transcriptome and genome sequencing associated with museum vouchers?

11 of the 18 new genomes and transcriptomes come from vouchered specimens at either Scripps (SIO) or the Paris Museum (MNHN). Voucher information is reported in Table S1.

Figure 1. Small specimens are hardly visible even after zooming (please check resolution).

Resolution is high in the original files.

Figure 2. (A) says "Favored topology" – could you please state to which analysis the phylogram corresponds to? In line 154: please, avoid the word "significant" unless referring to statistical tests. (D) why does the legend have gradients of green, blue and red? The meaning is not clear to me.

Corrected, clarified, and expanded. I hope all of this is clearer in the newer version.

Reviewer #2 (Recommendations for the authors):I found the text long in places and thus tedious to read. Particularly the introduction and the discussion. The intro could have a tighter narrative more focused on the discrepancies of the fossil record and divergence times, and on discrepancies in topology without the needed to review so much echinoid biology. The discussion appears too long. Too much effort is made on justifying the chronospace approach. This perhaps does not need to be justified at all beyond a sentence or two.

The issue of the length of the introduction and the extent to which echinoid background is developed seems to be a discrepancy between reviewers. We have decided not to modify its length, as we believe the present version is already a good compromise, as is suggested by Reviewer #1 as well. We have however reduced the length of the justification of chronospaces.

Because the clock model has such an impact, you should explore this further. PhyloBayes provides the ability to test for the various rate models using Bayes factors and you should try this. Because this analyses are computationally expensive, you can do them on a reduced amount of data. MCMCtree also allows you to test for various clock models with Bayes factors.

Given the computational burden of running a Bayes factor analysis on PhyloBayes, we had explored CorrTest instead, a different approach for selecting among competing clock models. The results of this were not reported before, but are now incorporated as Reviewer #1 also suggested this. Given that these methods did not eliminate uncertainty in terms of which clock model should be preferred, we have taken the position of exploring and reporting the sensitivity of results to all factors. All conclusions drawn in the manuscript regarding echinoderm diversification are robust even to the large effects introduced by the choice of clock models. We don’t feel the need to restrict the analyses or results to a subset of conditions when this would not modify our insights.

Additional points:The y-axis (Δ likelihood) of panel C in Figure 2 needs to be explained in the legend. What is the benchmark likelihood?

Panel C of Figure 2 is better explained now in the caption. There is no benchmark likelihood, values larger than 0 mean support for one topology, those below 0 for the alternative.

In the figure legends, the program/models used to infer the tree and times should be indicated. From the methods it's clear that many many different methods were tested, but from the main text and the figure legends is not clear what is being summarised in the figures.

Software are now mentioned in the captions of Figures 2 and 4.

Please remove the p-value for the multivariate analysis in line 198. This is not a replicated stochastic experiment. You are simply changing priors and substitution models and hence the posterior changes. This is a deterministic mapping between data/prior and posterior.

We agree with this comment and have removed the mention to p-values.

Panel C of Figure 4: please mention in the legend the scale of the x-axis tickmarks (10 My?).

Done.

Lines 359 to 367 have little substance. There are no figures in this paper showing the correlation structure among branch lengths. For example, figure 4 shows stacked posterior distributions, like those in previous works, so isn't this work a victim of its own criticism? There is indeed a correlation structure among branches and times, which is not shown, and which is discarded in the plots shown here. To emphasise this point, here is paper's text edited: "The sensitivity of inferred ages is commonly explored by running analyses under different settings and summarizing the results in tables or by stacking chronograms in order to visualize the relative position of nodes (see for example Figure 4C here and the supplementary material figures)."

We agree with the reviewer and have removed all of the text mentioned here, as well as other parts referring to the chronospace approach. Correlation plots were shown in a previous version of the manuscript, but we agree that these sentences do not add much after these have been removed.